# CAN VISUAL INPUT BE COMPRESSED? A VISUAL INPUT TOKEN COMPRESSION BENCHMARK FOR LARGE MULTIMODAL MODELS

## ABSTRACT

Large multimodal models (LMMs) often suffer from severe inference inefficiency due to the large number of visual tokens introduced by image encoders. While recent token compression methods, such as pruning and merging, have shown promise in reducing redundancy, their evaluation remains fragmented and inconsistent. In this work, we present **UniPruneBench**, a unified and extensible benchmark for visual token pruning in multimodal LLMs. UniPruneBench provides standardized protocols across six ability dimensions and ten datasets, covering ten representative compression algorithms and three families of LMMs (LLaVA-v1.5, Intern-VL3, and Qwen2.5-VL). Beyond task accuracy, it incorporates system-level metrics such as runtime and prefilling latency to provide a holistic view. Our experiments uncover several key findings: (1) random pruning is a surprisingly strong baseline, (2) no single method consistently outperforms others across scenarios, (3) pruning sensitivity varies significantly across tasks, with OCR being most vulnerable, and (4) pruning ratio is the dominant factor governing performance degradation. We believe UniPruneBench will serve as a reliable foundation for future research on efficient multimodal modeling.

## 1 INTRODUCTION

Recently, large multimodal models (LMMs) have achieved remarkable progress across a wide range of multimodal tasks. These models are typically built upon pre-trained large language models (LLMs) by integrating visual encoders (e.g., CLIP (Radford et al., 2021), CoCa (Yu et al., 2022)) and lightweight adapter modules. The visual encoders transform images into sequences of visual tokens, while the adapters bridge these visual representations with the textual space, enabling seamless multimodal understanding and reasoning. Representative LMMs follow two main adapter paradigms: BLIP-style models that rely on cross-modal attention and LLaVA-style models that concatenate ViT patch tokens into the LLM context with MLPl layers. These approaches, exemplified by LLaVA (Liu et al., 2023a), Qwen-VL (Yang et al., 2024), and Intern-VL (Zhu et al., 2025), have achieved strong performance in visual question answering (VQA), grounding, and multimodal reasoning.

However, incorporating visual inputs into LMMs inevitably introduces a large number of visual tokens, leading to substantial redundancy and creating a strong demand for faster inference (Vasu et al., 2025; Wen et al., 2025a). Unlike text tokens, which are semantically dense, visual tokens are often redundant and highly correlated(Bi et al., 2025b). Directly appending hundreds of tokens per image leads to steep increases in computation, memory usage, and inference latency, posing severe bottlenecks for real-time and large-scale deployment. Moreover, for many vision-language tasks such as VQA, it is unnecessary to process the entire image. Only a subset of task-relevant regions needs to be considered, further highlighting the inefficiency of uniform dense tokenization.

To address this, visual token compression has emerged as a promising direction. Recent work explores token pruning and merging to reduce redundancy while preserving essential semantics. For example, FastV (Vasu et al., 2025) prunes tokens with low attention scores, PyramidDrop (Xing et al., 2024) shrinks sequences in a layer-wise schedule, AdaptInfer (Zhang et al., 2025a) uses a dynamic text-guided mechanism to reuse hierarchical text-to-text attention maps to construct a soft prior for token importance, Recoverable Compression (Chen et al., 2025) restores filtered key tokens

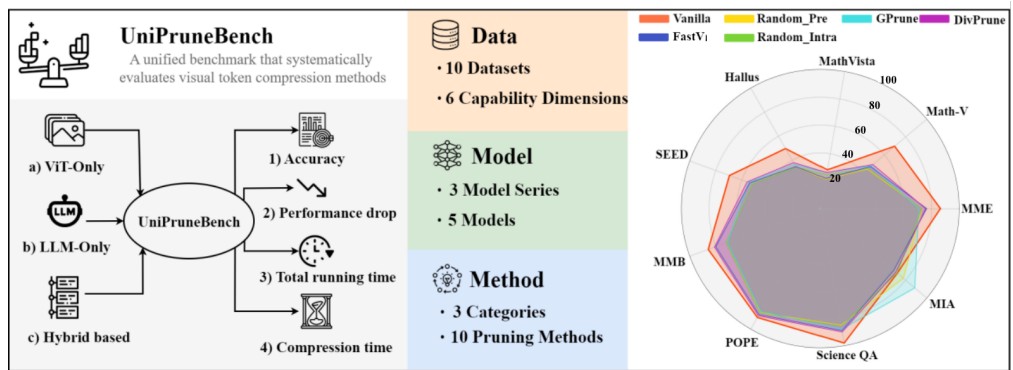

Figure 1: Overview of UniPruneBench, along with experimental results for representative pruning methods across various data scenarios.

related to the text based on text-visual similarity, DCP (Jiang et al., 2025a) selects key tokens related to the text through a text-aware computation module and preserves visual structural information through an image-aware computation module, HoloV (Zou et al., 2025) generates a composite score by combining intra-modal semantic diversity with visual attention saliency, and GridPrune (Duan et al., 2025) dynamically allocates token budgets to image grid regions based on text-conditioned correlation. Such methods reduce computation and memory and are compatible with existing LMMs due to their ability to process variable-length inputs.

Despite these advances, existing token compression methods lack a fair and systematic evaluation: **1) Limited coverage of methods, model series, and downstream tasks:** Most prior work evaluates only a small subset of compression algorithms on a narrow set of datasets, preventing a comprehensive understanding across different downstream task scenarios. **2) Absence of standardized evaluation protocols:** Current studies adopt heterogeneous frameworks, such as LLaVA native eval, LMMS-Eval (Zhang et al., 2024), and VLMEvalKit (Duan et al., 2024b), with inconsistent prompt templates, scoring metrics, and token retention ratios. These inconsistencies make it difficult to reliably compare methods and reproduce reported results. **3) Neglect of system-level metrics and low modularity:** Evaluations typically focus on task accuracy while ignoring important metrics such as runtime and end-to-end latency. In addition, many pruning implementations are tightly coupled to specific architectures, limiting their flexibility and making it challenging to extend them to emerging multimodal models.

These limitations highlight the need for a **unified, extensible, and user-friendly benchmark** to enable fair and reproducible evaluation of token compression methods for multimodal LLMs. To achieve this, in this paper, we introduce the **Uni**fied **Vis**ual Token **Prun**ing **Bench**mark (**UniPruneBench**), a benchmark designed to systematically evaluate plug-and-play visual token compression algorithms. As shown in Fig 1, UniPruneBench is a standardized evaluation benchmark that offers a fair and unified platform for comparing visual token pruning techniques. In addition, it provides a modular and user-friendly interface that decouples pruning logic from model architecture, enabling seamless integration with various LMMs.

Specifically, UniPruneBench provides **(1)** a diverse and challenging benchmark spanning **six ability dimensions** (e.g., comprehensive understanding, OCR, mathematical reasoning, and hallucination) **across ten datasets**. **(2)** It categorizes existing plug-and-play token compression methods **into three types: ViT-only, LLM-only, and hybrid**, based on where token pruning is applied, and offers comprehensive evaluations of **ten representative algorithms**. Besides, **(3)** it conducts experiments on **three series of large multimodal models: LLaVA-v1.5, Intern-VL3, and Qwen2.5-VL**. In addition to measuring performance drop, **(4)** the benchmark also reports system-level metrics, **including total running time, prefilling time**, providing a holistic view of both accuracy and efficiency.

Through extensive experiments, UniPruneBench reveals key observations: **(1). Random pruning is a surprisingly strong baseline.** Despite its simplicity, random pruning often surpasses existing designed strategies, highlighting the need for stronger baselines; **(2). No single method dominates across scenarios.** Different methods excel under different models, pruning ratios, and datasets; **(3). Task sensitivity varies significantly.** Instruction-following tasks remain robust, while OCR benchmarks suffer the most severe degradation; **(4). Pruning ratio drives accuracy-efficiency**

**trade-offs.** Light pruning incurs only moderate drops, whereas aggressive pruning sharply degrades performance; **(5). These trends are consistent across all three models, indicating generality.**

## 2 RELATED WORK

### 2.1 LARGE MULTIMODAL MODEL

Large Multimodal Models (LMMs) have brought significant breakthroughs in integrating vision and language, enabling models to perform complex cross-modal understanding and reasoning tasks. A typical end-to-end LMM consists of three major components: a language encoder, a vision encoder, and a cross-modal interaction module (Caffagni et al., 2024). The language encoder is usually adapted from large language models like LLaMA (Grattafiori et al., 2024; Touvron et al., 2023) and Qwen (Yang et al., 2024), while the vision encoder often adopts architectures such as Vision Transformer (ViT) (Dosovitskiy et al., 2020). The cross-modality module connects the two modalities, allowing language models to process visual inputs effectively.

Based on this architecture, various LMMs have been developed with different design choices and training strategies. For instance, Qwen2.5-VL (Bai et al., 2025) introduces a visual receptor and follows a structured multi-stage training process. Intern-VL3 (Chen et al., 2024b) adopts joint multimodal pretraining across large-scale datasets, while LLaVA (Liu et al., 2023a) and its successor LLaVA-OneVision (Li et al., 2025) focus on enhancing visual grounding and reasoning through task-aligned training with MLP layer. These approaches collectively push the limits of vision-language alignment, leading to strong performance across a variety of multimodal benchmarks (Kil et al., 2024; Huang & Zhang, 2024). In addition, recent work explores the use of LLM agents equipped with visual tools (Gao et al., 2024; Fan et al., 2024; Gupta & Kembhavi, 2023; Bi et al., 2025c) to handle more dynamic and interactive multimodal tasks. However, such agent-based methods go beyond the scope of this paper, which focuses on the architectural and training advances of the agent framework.

### 2.2 VISUAL TOKEN COMPRESSION BENCHMARK.

Few benchmarks have been proposed for this task. The most relevant prior analysis work proposed in (Wen et al., 2025a), which evaluates only four token-pruning baselines, namely FastV, Sparse-VLM, random, and pooling. And it does not provide source code or reproducible scripts. In contrast, our work implements 10 state-of-the-art pruning algorithms and will release all implementation details publicly. Moreover, we evaluate these methods across a wider range of downstream tasks and metrics, providing a more comprehensive and reproducible benchmark.

## 3 UNIPRUNEBENCH

The UniPruneBench comprises six evaluation dimensions spanning ten datasets, each released under permissive licenses that permit research use. In addition, it benchmarks ten representative methods across three categories and covers five representative LMMs from three different model families.

### 3.1 VISUAL TOKEN COMPRESSION METHODS

LMM pruning aims to reduce redundant tokens while preserving model performance. Especially, The number of visual tokens is usually tens to hundreds of times that of language tokens, and visual signals are inherently more sparse, thus needing to be pruned. As shown in Fig 2, Existing plug-and-play methods can be divided into three categories according to where to prune: **ViT-only method** (Bolya et al., 2023; Jiang et al., 2025b), **LLM-only method** (Wen et al., 2025b; Ye et al., 2025) , and **Hybrid based method** (Zhang et al., 2025b; Liu et al., 2024a). Besides, one very basic method, i.e., **random token pruning**, is also adopted as a strong baseline.

**ViT-only methods:** Token pruning in ViTs is achieved through two paradigms: token selection and token merging. For token selection, **DivPrune** (Alvar et al., 2025) formulates pruning as a subset selection problem that maximizes diversity, thus reducing redundancy while preserving representative information. **G-Prune** (Jiang et al., 2025b) iteratively updates importance scores via information propagation, retaining the most representative tokens from both foreground and back-

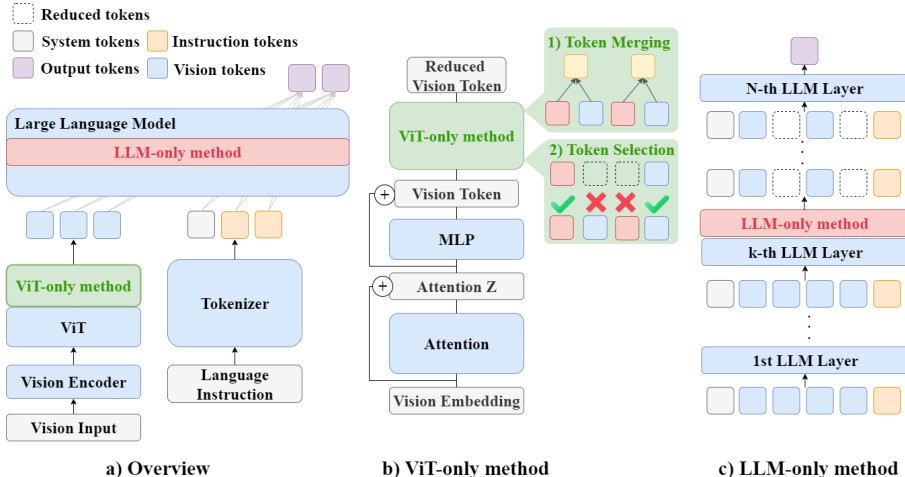

Figure 2: Taxonomy of visual token pruning Methods, including ViT-only, LLM-only, and hybrid.

ground regions. **LLaVA-PruMerge** (Shang et al., 2024) further optimizes token processing in the CLIP image encoder through an adaptive token merging strategy.

**LLM-only methods:** These approaches prune visual tokens within the LLM to reduce computation while maintaining performance. **FastV** (Chen et al., 2024a) makes an early attempt by discarding tokens after the second layer of LMMs. **VTW** (Lin et al., 2025) argues that tokens can be entirely removed once the model reaches sufficient depth. **DART** (Wen et al., 2025b) selects a small set of pivot tokens and removes others with high redundancy.

**Hybrid methods:** Multi-stage hybrid pruning combines strategies across different components of LMMs. **SparseVLM** uses a rank-based strategy to set sparsification ratios adaptively and recycles pruned tokens into compact representations (Zhang et al., 2025b). **MustDrop** evaluates token importance across visual encoding, prefill, and decode, applying stage-specific strategies to remove redundancy and reduce computation (Liu et al., 2024a).

### 3.2 DATASETS

To systematically assess the impact of pruning techniques on LMMs, we conduct experiments on ten benchmark datasets covering six capability dimensions: **(1) Comprehensive Evaluation**: MME (Fu et al., 2023), and MMBench (Liu et al., 2024b); **(2) Mathematical Reasoning**: MathVista (Lu et al., 2024), and Math-Vision (Wang et al., 2025); **(3) Optical Character Recognition**: SEED-Bench-2-Plus (Li et al., 2024), and OCRBench (Liu et al., 2023b); **(4) Instruction Following**: MIA-Bench (Qian et al., 2024); **(5) Multidisciplinary Knowledge**: ScienceQA (Lu et al., 2022); **(6) Hallucination**: POPE (Li et al., 2023), and HallusionBench (Guan et al., 2024). Our dataset selection follows the capability dimensions identified in MME-Survey (Fu et al., 2024), ensuring coverage of core competencies such as visual–language understanding, cross-modal reasoning, and instruction following, skills central to current LMM research and applications.

### 3.3 BASE MODEL

Following prior work (Zhang et al., 2025b; Liu et al., 2024a; Bi et al., 2025a), we evaluate five representative open-source LMMs from three model families, namely LLaVA-v1.5-7B, InternVL3-1B, InternVL3-8B, Qwen2.5-VL-3B and Qwen2.5-VL-7B. These models align vision and language by integrating advanced visual and textual components. Typically, a visual encoder (e.g., CLIP) processes image inputs, while a large language model (e.g., Qwen) handles textual inputs. The extracted features are then fused via Multilayer Perceptron (MLP) connectors, enabling effective multimodal reasoning and alignment. Previous studies typically conduct experiments on only one or two model families, whereas we evaluate models across all three families.

### 3.4 EVALUATION METRICS

We consider multiple metrics in this benchmark. **Accuracy** serves as a basic evaluation metric, complemented by **performance drop** measured before and after token compression. To reflect practical considerations, we also record **total running time**, the time required for the entire process,

Table 1: Performance comparison across different methods and benchmarks on LLaVA-v1.5-7B.

| Methods | Comprehensive | | | OCR | | Multidisciplinary | Hallucination | Avg. |
|---|---|---|---|---|---|---|---|---|
| | MME | MMB-cn | MMB-en | SEED | OCR-B | Science QA | POPE | |
| **LLaVA-v1.5-7B** | *Upper Bound: 576 Tokens (100%)* | | | | | | | |
| Vanilla | 48.1 | 43.5 | 63.6 | 38.8 | 30.2 | 68.2 | 80.1 | 53.2 |
| | *Retain Averaged 192 Tokens (↓66.7%)* | | | | | | | |
| **Random** | 44.3 ± 1.8 | 39.4 ± 2.1 | 57.1 ± 2.6 | 38.3 ± 2.2 | 21.6 ± 2.1 | 65.7 ± 1.3 | 82.1 ± 0.3 | 52.3 |
| | ↓3.0% | ↓4.4% | ↓5.3% | ↑4.5% | ↓17.9% | ↑0.0% | ↑5.1% | |
| **VTW** | 23.3 | 0.8 | 21.0 | 36.9 | 0.9 | 63.1 | 4.9 | 21.6 |
| | ↓51.6% | ↓98.2% | ↓66.9% | ↓4.9% | ↓97.0% | ↓7.5% | ↓93.9% | |
| **PruMerge** | 44.0 | 6.0 | 57.1 | 38.3 | 23.4 | 66.5 | 74.5 | 44.3 |
| | ↓8.6% | ↓86.3% | ↓10.2% | ↓1.3% | ↓22.5% | ↓2.5% | ↓6.9% | |
| **FastV** | 44.9 | 23.1 | 60.5 | 37.2 | 27.0 | 68.1 | 74.8 | 47.9 |
| | ↓6.7% | ↓46.9% | ↓4.8% | ↓4.2% | ↓10.6% | ↓0.2% | ↓6.6% | |
| **DivPrune** | **49.2** | 12.3 | 60.4 | 38.5 | 28.4 | 67.6 | **86.4** | 49.0 |
| | ↑2.2% | ↓71.8% | ↓5.0% | ↓0.9% | ↓6.0% | ↓0.9% | ↑7.9% | |
| **VisPrune** | 47.3 | 18.8 | 61.6 | 38.3 | 29.1 | 68.0 | 85.2 | 49.8 |
| | ↓1.7% | ↓56.9% | ↓3.1% | ↓1.5% | ↓3.6% | ↓0.2% | ↑6.4% | |
| **DART** | 46.8 | 25.8 | 61.7 | 37.8 | 28.3 | 66.3 | 83.2 | 50.0 |
| | ↓2.7% | ↓40.7% | ↓3.0% | ↓2.7% | ↓6.3% | ↓2.8% | ↑4.0% | |
| **MustDrop** | 47.8 | 41.3 | 61.0 | 37.9 | 28.9 | 67.6 | 80.1 | 52.1 |
| | ↓0.8% | ↓5.2% | ↓4.1% | ↓2.2% | ↓4.3% | ↓0.8% | ↑0.1% | |
| **SparseVLM** | 47.7 | **44.0** | **61.7** | 39.7 | 28.1 | 67.4 | 80.6 | **52.7** |
| | ↓1.0% | ↑1.2% | ↓2.8% | ↑2.3% | ↓7.0% | ↓1.1% | ↑0.6% | |
| | *Retain Averaged 128 Tokens (↓77.8%)* | | | | | | | |
| **Random** | 45.2 ± 0.3 | 37.1 ± 0.9 | 54.2 ± 1.2 | 38.1 ± 1.0 | 20.2 ± 2.8 | 64.0 ± 0.3 | 81.6 ± 0.3 | 50.4 |
| | ↓5.7% | ↓10.1% | ↓9.1% | ↑2.7% | ↓26.5% | ↓1.7% | ↑2.0% | |
| **VTW** | 24.7 | 1.1 | 23.5 | 36.9 | 1.0 | 64.5 | 0.0 | 21.7 |
| | ↓48.7% | ↓97.4% | ↓63.1% | ↓4.9% | ↓96.7% | ↓5.4% | ↓100.0% | |
| **PruMerge** | 41.5 | 4.8 | 55.4 | 38.6 | 23.7 | 67.6 | 69.8 | 43.1 |
| | ↓13.8% | ↓89.1% | ↓12.8% | ↓0.6% | ↓21.5% | ↓0.9% | ↓12.9% | |
| **FastV** | 41.9 | 20.7 | 57.8 | 36.9 | 25.4 | **68.3** | 67.5 | 45.5 |
| | ↓13.0% | ↓52.5% | ↓9.1% | ↓5.0% | ↓15.9% | ↑0.2% | ↓15.6% | |
| **DivPrune** | **48.9** | 9.0 | 59.5 | 39.1 | 27.9 | 67.5 | **86.4** | 48.3 |
| | ↑1.7% | ↓79.3% | ↓6.4% | ↑0.8% | ↓7.6% | ↓1.0% | ↑7.9% | |
| **VisPrune** | 47.9 | 14.5 | 60.3 | 38.3 | 29.4 | 67.8 | 83.9 | 48.9 |
| | ↓0.5% | ↓66.8% | ↓5.2% | ↓1.5% | ↓2.6% | ↓0.6% | ↑4.8% | |
| **DART** | 46.1 | 22.4 | 61.0 | 37.6 | 26.7 | 67.1 | 79.9 | 48.7 |
| | ↓4.3% | ↓48.5% | ↓4.1% | ↓3.0% | ↓11.6% | ↓1.6% | ↓0.2% | |
| **MustDrop** | 47.4 | 41.7 | 61.1 | 39.6 | **29.6** | 67.5 | 78.9 | 52.3 |
| | ↓1.5% | ↓4.2% | ↓3.8% | ↑1.9% | ↓2.0% | ↓0.9% | ↓1.5% | |
| **SparseVLM** | **48.9** | **48.2** | **62.4** | **39.9** | 24.9 | 67.4 | 83.1 | **53.5** |
| | ↑1.5% | ↑10.7% | ↓1.8% | ↑2.8% | ↓17.5% | ↓1.2% | ↑3.7% | |
| | *Retain Averaged 64 Tokens (↓88.9%)* | | | | | | | |
| **Random** | 42.2 ± 0.2 | 7.3 ± 3.6 | 52.4 ± 1.1 | 36.5 ± 2.7 | 18.4 ± 2.5 | 62.2 ± 0.9 | 74.9 ± 0.2 | 42.6 |
| | ↓12.1% | ↓88.7% | ↓17.0% | ↓3.4% | ↓36.8% | ↓2.9% | ↓6.2% | |
| **VTW** | 25.0 | 4.4 | 50.0 | 38.3 | 1.4 | 65.7 | 9.2 | 27.7 |
| | ↓48.0% | ↓89.9% | ↓21.4% | ↓1.5% | ↓95.4% | ↓3.6% | ↓88.6% | |
| **PruMerge** | 41.2 | 4.5 | 53.1 | 38.8 | 22.8 | 67.8 | 65.1 | 41.9 |
| | ↓14.3% | ↓89.7% | ↓16.5% | ↓0.1% | ↓24.5% | ↓0.5% | ↓18.7% | |
| **FastV** | 32.2 | 12.7 | 45.8 | 36.2 | 16.8 | 67.2 | 51.3 | 37.5 |
| | ↓33.0% | ↓70.8% | ↓27.9% | ↓6.8% | ↓44.4% | ↓1.5% | ↓35.9% | |
| **DivPrune** | **48.1** | 6.1 | 57.9 | **39.0** | 26.9 | 65.9 | **85.3** | **47.0** |
| | ↑0.0% | ↓85.9% | ↓8.9% | ↑0.3% | ↓10.9% | ↓3.3% | ↑6.5% | |
| **VisPrune** | 47.8 | 7.5 | 58.2 | 38.6 | **28.1** | 67.5 | 80.7 | 46.9 |
| | ↓0.7% | ↓82.7% | ↓8.4% | ↓0.7% | ↓7.0% | ↓1.0% | ↑0.8% | |
| **DART** | 42.8 | **17.4** | 57.4 | 37.6 | 23.4 | 67.9 | 71.0 | 45.4 |
| | ↓11.1% | ↓60.0% | ↓9.7% | ↓3.0% | ↓22.5% | ↓0.4% | ↓11.3% | |
| **MustDrop** | 43.9 | 13.2 | 57.4 | 38.0 | 24.8 | **68.5** | 67.2 | 44.7 |
| | ↓8.7% | ↓69.8% | ↓9.7% | ↓2.1% | ↓17.9% | ↑0.4% | ↓16.1% | |
| **SparseVLM** | 45.5 | 15.7 | **58.8** | 38.1 | 16.7 | 67.8 | 76.8 | 45.6 |
| | ↓5.5% | ↓63.8% | ↓7.5% | ↓1.9% | ↓44.7% | ↓0.6% | ↓4.0% | |

and **compression strategy time**, the time spent solely on token pruning, and **prefilling-phase time**, the forward pass that processes image and text tokens before the first decoding step.

## 4 EXPERIMENTS

### 4.1 IMPLEMENTATION DETAILS

We implemented UniPruneBench in PyTorch and conducted all experiments on NVIDIA A100 GPUs. For benchmark execution across baselines and models, we employed the open-source toolkit VLMEvalKit (Duan et al., 2024a). Unless otherwise specified, all results are reported with a batch

Table 2: Performance comparison across different methods and benchmarks on InternVL3-8B.

| Methods | Comprehensive | | OCR | Multidisciplinary | Hallucination | | Mathematical | | Instruction | Avg. |
|---|---|---|---|---|---|---|---|---|---|---|
| | MME | MMB-en | SEED | Science QA | POPE | Hallus | Math-V | MathVista | MIA | |
| **InternVL3-8B** | | | | *Upper Bound, 100% Tokens* (100%) | | | | | | |
| Vanilla | 86.44 | 85.80 | 69.52 | 98.07 | 90.33 | 49.80 | 69.70 | 28.29 | 72.22 | 70.02 |
| | | | | *Retain Averaged 33.3% Tokens* (↓ 66.7%) | | | | | | |
| **Random-Pre** | 77.21 | 78.79 | 56.65 | 90.00 | 88.58 | 37.25 | 50.00 | 22.70 | 80.93 | 64.68 |
| | ↓10.7% | ↓8.2% | ↓18.5% | ↓8.2% | ↓1.9% | ↓25.2% | ↓28.3% | ↓19.8% | ↑12.1% | |
| **Random-Intra** | 81.04 | 82.00 | 60.00 | 94.00 | 89.70 | 41.09 | 53.90 | 27.63 | 74.50 | 67.10 |
| | ↓6.2% | ↓4.4% | ↓13.7% | ↓4.2% | ↓0.7% | ↓17.5% | ↓22.7% | ↓2.3% | ↑3.2% | |
| **FastV** | 80.60 | 83.00 | 58.00 | 92.00 | 90.21 | 42.95 | 49.80 | 24.34 | 74.74 | 66.18 |
| | ↓6.8% | ↓3.3% | ↓16.6% | ↓6.2% | ↓0.1% | ↓13.8% | ↓28.6% | ↓14.0% | ↑3.5% | |
| **DivPrune** | 81.76 | **84.00** | **63.00** | 95.00 | 90.09 | 44.08 | 60.50 | **26.97** | 79.66 | 69.45 |
| | ↓5.4% | ↓2.1% | ↓9.4% | ↓3.1% | ↓0.3% | ↓11.5% | ↓13.2% | ↓4.7% | ↑10.3% | |
| **GPrune** | **82.62** | 83.00 | 62.00 | **96.00** | 90.12 | 46.91 | 64.60 | 26.97 | **81.61** | **70.43** |
| | ↓4.4% | ↓3.3% | ↓10.8% | ↓2.1% | ↓0.2% | ↓5.8% | ↓7.3% | ↓4.7% | ↑13.0% | |
| | | | | *Retain Averaged 22.2% Tokens* (↓ 77.8%) | | | | | | |
| **Random-Pre** | 77.30 | 77.00 | 55.00 | 88.00 | 88.08 | 35.52 | 47.60 | 21.38 | 77.34 | 62.91 |
| | ↓10.6% | ↓10.2% | ↓20.9% | ↓10.2% | ↓2.5% | ↓28.7% | ↓31.7% | ↓24.4% | ↑7.1% | |
| **Random-Intra** | 78.13 | 79.00 | 57.00 | 91.00 | 88.93 | 38.24 | 48.10 | 22.04 | 72.69 | 63.90 |
| | ↓9.6% | ↓7.9% | ↓18.0% | ↓7.1% | ↓1.6% | ↓23.2% | ↓31.0% | ↓22.1% | ↑0.6% | |
| **FastV** | 79.40 | 83.00 | 56.00 | 91.00 | 89.37 | 38.29 | 49.70 | **26.32** | 72.20 | 65.03 |
| | ↓8.1% | ↓3.3% | ↓19.4% | ↓7.1% | ↓1.1% | ↓23.1% | ↓28.7% | ↓7.0% | ↓0.01% | |
| **GPrune** | 78.97 | 79.00 | **60.00** | **94.00** | 89.33 | 43.70 | 55.50 | 24.01 | 74.70 | 66.58 |
| | ↓8.6% | ↓7.9% | ↓13.7% | ↓4.1% | ↓1.1% | ↓12.3% | ↓20.4% | ↓15.1% | ↑3.4% | |
| **DivPrune** | **80.25** | **83.00** | **60.00** | 93.00 | **90.18** | 42.64 | **56.00** | 23.36 | **79.82** | **67.58** |
| | ↓7.2% | ↓3.3% | ↓13.7% | ↓5.1% | ↓0.2% | ↓14.4% | ↓19.7% | ↓17.4% | ↑10.5% | |
| | | | | *Retain Averaged 11.1% Tokens* (↓ 88.9%) | | | | | | |
| **Random-Pre** | 73.11 | 72.00 | 52.00 | 85.00 | 86.32 | 34.72 | 44.00 | 21.38 | 77.93 | 60.72 |
| | ↓15.4% | ↓16.1% | ↓25.2% | ↓13.3% | ↓4.4% | ↓30.3% | ↓36.9% | ↓24.4% | ↑7.8% | |
| **Random-Intra** | 72.97 | 72.00 | 52.00 | 86.00 | 86.61 | 34.47 | 44.90 | 24.67 | 71.97 | 60.73 |
| | ↓15.6% | ↓16.1% | ↓25.2% | ↓12.2% | ↓4.1% | ↓30.8% | ↓35.6% | ↓12.8% | ↓0.3% | |
| **FastV** | **76.49** | 80.00 | 54.00 | 89.00 | 88.00 | 34.88 | 47.00 | 22.04 | 68.97 | 62.26 |
| | ↓11.5% | ↓6.8% | ↓22.3% | ↓9.2% | ↓2.6% | ↓30.0% | ↓32.6% | ↓22.1% | ↓4.5% | |
| **GPrune** | 70.82 | 71.00 | 55.00 | 88.00 | 85.35 | 36.68 | 47.90 | **26.32** | **88.71** | 63.31 |
| | ↓18.1% | ↓17.3% | ↓20.9% | ↓10.2% | ↓5.5% | ↓26.3% | ↓31.2% | ↓7.0% | ↑22.8% | |
| **DivPrune** | 75.79 | **81.00** | **56.00** | **90.00** | 88.95 | 38.13 | 49.20 | 26.32 | 70.97 | **64.04** |
| | ↓12.3% | ↓5.6% | ↓19.4% | ↓8.2% | ↓1.5% | ↓23.4% | ↓29.4% | ↓7.0% | ↓1.8% | |

size of 1. We evaluate performance under different pruning ratios, ensuring that pruned models maintain sufficiently high accuracy for meaningful comparison with baselines. To enable fair comparisons across benchmarks of varying scales, we report both average accuracy and relative performance. To ensure a fair comparison across all Intra-LLM pruning strategies, we fix the pruning location inside the large model at layer K = 2 for every method. For clarity, Random-Pre denotes uniform random dropping applied to visual tokens before they enter the LLM (Pre-LLM stage), whereas Random-Intra performs the same stochastic removal inside the LLM (Intra-LLM stage); both retain exactly the target sparsity yet introduce no learned importance bias. For task performance metrics, we have normalized to 0-100 for MME to align with other datasets, with higher values indicating better results, while for runtime measurements, lower values are preferable. To simplify presentation, we adopt the following dataset abbreviations when reporting results: MMBench as MMB, Math-Vision as Math-V, SEEDBench-2-Plus as SEED, OCRBench as OCR-B, MIA-Bench as MIA, and HallusionBench as Hallus.

## 4.2 MAIN RESULTS

The comparison results of different methods are shown in Table 1, Table 2, and Table 3. Based on these results, several key findings emerge:

**1. Random pruning remains a surprisingly strong baseline.** Random pruning consistently outperforms several well-designed methods, such as GPrune, VTW, and PruMerge. For instance, On LLaVA-v1.5-7B, six out of eight perform worse than random pruning at 66.7% and 77.8% pruning

Table 3: Performance comparison across different methods and benchmarks on Qwen2.5-VL-7B.

| Methods | Comprehensive | | OCR | | Multidisciplinary | Hallucination | | Mathematical | | Instruction | Avg. |
|---|---|---|---|---|---|---|---|---|---|---|---|
| | MME | MMB-en | SEED | OCR-B | Science QA | POPE | Hallus | Math-V | MathVista | MIA | |
| **Qwen2.5-VL-7B** | | | | | *Upper Bound: 100% Token (100%)* | | | | | | |
| Vanilla | 82.5 | 79.8 | 69.6 | 78.3 | 89.0 | 87.5 | 47.5 | 24.3 | 63.9 | 70.2 | 69.3 |
| | | | | | *Retain Averaged 33.3% Tokens (↓66.7%)* | | | | | | |
| **FastV** | 69.4 | 74.1 | 59.0 | 50.5 | **80.0** | 83.7 | 37.9 | 9.34 | 40.2 | 60.9 | 56.5 |
| | ↓15.9% | ↓7.1% | ↓15.2% | ↓35.5% | ↓10.1% | ↓4.3% | ↓20.2% | ↓61.6% | ↓37.1% | ↓13.2% | |
| **Random-Intra** | 67.7 | 74.7 | 58.0 | 53.3 | 79.0 | 81.9 | 38.6 | 9.51 | 41.0 | 61.9 | 56.6 |
| | ↓17.9% | ↓6.4% | ↓16.7% | ↓31.9% | ↓11.2% | ↓6.4% | ↓18.7% | ↓60.9% | ↓35.8% | ↓11.8% | |
| **GPrune** | 68.6 | 71.2 | 57.0 | 54.9 | **80.0** | 84.4 | 37.8 | 9.8 | 47.8 | 60.10 | 57.2 |
| | ↓16.8% | ↓10.8% | ↓18.1% | ↓29.9% | ↓10.1% | ↓3.5% | ↓20.4% | ↓59.7% | ↓25.2% | ↓14.4% | |
| **Random-Pre** | 71.9 | 71.2 | 57.0 | 54.9 | **80.0** | 84.4 | 37.8 | 9.41 | 45.7 | **62.9** | 57.5 |
| | ↓12.8% | ↓10.8% | ↓18.1% | ↓29.9% | ↓10.1% | ↓3.5% | ↓20.4% | ↓61.3% | ↓28.5% | ↓10.4% | |
| **DART** | 71.8 | **76.5** | 54.0 | 58.2 | **80.0** | 81.9 | **41.6** | 9.1 | 43.3 | 61.69 | 57.8 |
| | ↓13.0% | ↓4.1% | ↓22.4% | ↓25.7% | ↓10.1% | ↓6.4% | ↓12.4% | ↓62.6% | ↓32.2% | ↓12.1% | |
| **DivPrune** | **72.3** | 73.7 | **63.0** | 65.4 | 79.0 | 84.3 | **41.6** | 9.57 | **48.3** | 61.5 | **59.8** |
| | ↓12.4% | ↓7.6% | ↓9.5% | ↓16.5% | ↓11.2% | ↓3.7% | ↓12.4% | ↓60.6% | ↓24.4% | ↓12.4% | |
| | | | | | *Retain Averaged 22.2% Tokens (↓77.8%)* | | | | | | |
| **GPrune** | 59.8 | 65.1 | 52.0 | 38.1 | 78.0 | 80.6 | 31.5 | **9.8** | 47.5 | 60.71 | 52.3 |
| | ↓27.5% | ↓18.4% | ↓25.3% | ↓51.3% | ↓12.4% | ↓7.9% | ↓33.7% | ↓59.7% | ↓25.7% | ↓13.5% | |
| **FastV** | 66.4 | 70.6 | 55.0 | 36.4 | 78.0 | 80.7 | 35.6 | 9.21 | 38.1 | 60.3 | 53.0 |
| | ↓19.5% | ↓11.5% | ↓21.0% | ↓53.5% | ↓12.4% | ↓7.8% | ↓25.1% | ↓62.1% | ↓40.4% | ↓14.1% | |
| **Random-Intra** | 64.7 | 71.0 | 53.0 | 45.6 | 78.0 | 80.1 | 36.2 | 8.55 | 38.0 | 62.3 | 53.7 |
| | ↓21.6% | ↓11.0% | ↓23.9% | ↓41.8% | ↓12.4% | ↓8.5% | ↓23.8% | ↓64.8% | ↓40.5% | ↓11.3% | |
| **Random-Pre** | 67.9 | 69.3 | 54.0 | 45.6 | 78.0 | 79.9 | 34.2 | 9.67 | 45.5 | **63.0** | 54.7 |
| | ↓17.7% | ↓13.2% | ↓22.4% | ↓41.8% | ↓12.4% | ↓8.7% | ↓28.8% | ↓60.2% | ↓28.8% | ↓10.3% | |
| **DART** | 68.4 | **74.5** | 50.0 | 50.7 | **80.0** | 79.9 | **40.0** | 9.6 | 40.1 | 60.83 | 55.4 |
| | ↓17.1% | ↓6.6% | ↓28.2% | ↓35.3% | ↓10.1% | ↓8.7% | ↓15.8% | ↓60.5% | ↓37.2% | ↓13.4% | |
| **DivPrune** | **70.0** | 73.0 | **59.0** | 57.5 | 79.0 | **82.8** | 37.1 | 9.8 | **47.7** | 61.3 | **57.7** |
| | ↓15.2% | ↓8.5% | ↓15.2% | ↓26.6% | ↓11.2% | ↓5.4% | ↓21.9% | ↓59.7% | ↓25.4% | ↓12.7% | |
| | | | | | *Retain Averaged 11.1% Tokens (↓88.9%)* | | | | | | |
| **FastV** | 51.4 | 53.2 | 47.0 | 18.9 | 74.0 | 69.2 | 29.3 | 8.98 | 30.3 | 58.2 | 44.0 |
| | ↓37.7% | ↓33.3% | ↓32.5% | ↓75.9% | ↓16.9% | ↓21.0% | ↓38.3% | ↓63.0% | ↓52.6% | ↓17.1% | |
| **GPrune** | 49.9 | 53.7 | 46.0 | 16.4 | 71.0 | 70.0 | 24.4 | 9.24 | 47.4 | 60.3 | 44.8 |
| | ↓39.5% | ↓32.7% | ↓33.9% | ↓79.1% | ↓20.2% | ↓20.0% | ↓48.6% | ↓62.0% | ↓25.8% | ↓14.1% | |
| **Random-Intra** | 62.5 | 68.3 | 47.0 | 31.1 | 75.0 | 74.5 | 32.4 | 8.09 | 31.9 | 61.3 | 49.2 |
| | ↓24.2% | ↓14.4% | ↓32.5% | ↓60.3% | ↓15.7% | ↓14.9% | ↓31.8% | ↓66.7% | ↓50.1% | ↓12.7% | |
| **Random-Pre** | 64.1 | 64.8 | 48.0 | 31.7 | 73.0 | 73.4 | 27.7 | 9.80 | 45.3 | **62.7** | 50.1 |
| | ↓22.3% | ↓18.8% | ↓31.0% | ↓59.5% | ↓18.0% | ↓16.1% | ↓41.7% | ↓59.7% | ↓29.1% | ↓10.7% | |
| **DART** | 62.2 | **70.7** | 46.0 | 37.7 | **79.0** | 73.2 | **34.8** | **9.90** | 33.9 | 62.6 | 51.0 |
| | ↓24.6% | ↓11.4% | ↓33.9% | ↓51.9% | ↓11.2% | ↓16.3% | ↓26.7% | ↓59.3% | ↓46.9% | ↓10.8% | |
| **DivPrune** | **64.6** | 68.6 | **51.0** | 40.8 | 76.0 | **78.3** | 30.8 | 8.95 | **47.9** | 62.22 | **52.9** |
| | ↓21.7% | ↓14.0% | ↓26.7% | ↓47.9% | ↓14.6% | ↓10.5% | ↓35.2% | ↓63.2% | ↓25.0% | ↓11.4% | |

ratios. This unexpected result highlights the limitation of current designs and suggests that more effective pruning strategies are needed beyond naive baselines.

**2. No single method achieves universal superiority.** No approach dominates across all models and pruning ratios. DivPrune achieves the best results on both Qwen2.5-VL-7B and InternVL3-8B under all ratios. However, on LLaVA-v1.5-7B, SparseVLM surpasses DivPrune under light pruning ratios, while DivPrune regains superiority under more aggressive pruning. This indicates that performance strongly depends on both the model architecture and the pruning level.

**3. Hybrid-based methods demonstrate strong overall performance.** Among the three categories of methods, hybrid-based approaches achieve the best results on LLaVA-v1.5-7B at the 77.8% and 66.7% pruning ratios, though they perform worse at the 88.9% ratio. On InternVL3-8B and Qwen2.5-VL-7B, ViT-only methods (e.g., DivPrune) consistently outperform LLM-only methods, suggesting that vision-side pruning is more effective than language-side pruning.

**4. Task-level sensitivity varies: instruction following improves, while OCR degrades severely.** Most benchmarks show accuracy degradation as pruning intensifies. However, instruction-following tasks (e.g., MIA) exhibit improvements in some cases. For example, on InternVL3-8B, DivPrune raises accuracy from 72.22% to 79.82%. We hypothesize that pruning increases the relative weight of textual inputs, thereby enhancing instruction adherence. In contrast, OCR tasks are highly sensitive to pruning: as more visual tokens are removed, crucial details are lost, leading to rapid performance decline.

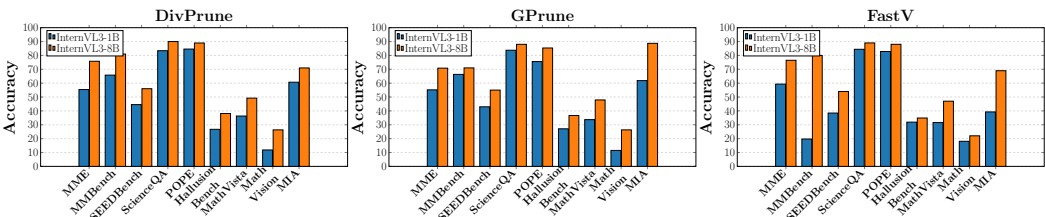

Figure 3: Performance Comparison of different model sizes for InternVL3 at 88.9% pruning rate.

**5. Higher pruning ratios induce sharper performance loss.** Light pruning leads to moderate degradation, while aggressive pruning causes substantial drops. For example, on Qwen2.5-VL-7B, the average accuracy decreases from 57.5% at 33% tokens to 50.1% at 11% tokens under random pruning. Similarly, on InternVL3-8B, DivPrune maintains 67.58% at 22% tokens but falls to 64.04% at 11% tokens. Notably, DivPrune consistently achieves the best results under the highest pruning ratio (88.9%), showing stronger robustness in extreme scenarios.

**6. Consistent cross-model trends.** Despite architectural differences, all three models exhibit similar behaviors: random pruning is unexpectedly competitive, OCR tasks are highly fragile, instruction-following tasks remain robust, and no single method dominates universally.

### 4.3 ANALYSIS AND DISCUSSION

**Influence of model size** To investigate the sensitivity of token compression techniques to model scale, we evaluate three representative methods, DivPrune, GPrune, and FastV, across two variants of InternVL: InternVL3-1B (small) and InternVL3-8B (large). As shown in Fig 3, scaling up the base model consistently yields significant accuracy gains across nearly all benchmarks under all compression methods, confirming that larger models retain more semantic capacity even after token reduction. The results indicate that larger architectures provide greater robustness to token reduction, suggesting that compression strategies should be evaluated across scales rather than in isolation.

**Running time** Considering real-world scenarios, we also evaluate the running time of different pruning methods. We profile three nested intervals: **Total time**, the elapsed time to finish the entire dataset; **Prefill time**, the single encoder forward pass that computes keys and values for all visual and textual tokens before any decoding starts, a phase that is compute-bound for the large model; and **Method time**, the GPU milliseconds spent only on the compression subroutine (token scoring, selection and tensor re-layout). All measurements were collected on an NVIDIA A100-40 GB GPU with batch size = 1 and three independent runs. All reported methods correspond to a uniform pruning rate of 88.9% on the MME benchmark. The results in Table 4 show that the last component never exceeds 0.5s, less than 0.12 % of the corresponding total. So the cost of importance estimation is negligible. Pruning therefore exerts its effect entirely within the prefill: DivPrune and GPrune shorten it from 320 s to 185 s and 167 s, delivering 1.73–1.92× encoder acceleration and an overall 1.62–1.68× end-to-end speed-up versus the vanilla model.

Table 4: The running time comparison of different methods on InternVL3-8B

| Methods | Total time (sec) | Prefill time (sec) | Method time (sec) |
| --- | --- | --- | --- |
| Vallania | 761.00 | 320.00 | 0.00 |
| Random-pre | 491.00 | 201.00 | **0.11** |
| Random-intra | 481.00 | 209.00 | 0.12 |
| Fastv | 497.00 | 212.00 | 0.33 |
| DivPrune | 469.00 | 185.00 | 0.32 |
| GPrune | **454.00** | **167.00** | 0.47 |

**Combination of different pruning strategies** To examine whether the same overall sparsity should be applied in one step or decomposed, we fix the global pruning ratio at 88.9% and realize it through two design choices: **Single-stage**, a single 88.9% drop executed either before the LLM (Pre-LLM) or inside the LLM (Intra-LLM). **Two-stage**, a 66.7% Pre-LLM pruning followed by 66.7% Intra-LLM pruning, giving the same compound retention. Contrary to the "more-is-better" intuition, Table 5

Table 5: **Performance comparison for the combination of different pruning strategies on InternVL3-8B.** All methods achieve 88.9% pruning rate. Mixed methods use two-stage pruning.

| Pre-LLM | Intra-LLM | Comprehensive | | OCR | Multidisciplinary | Hallucination | | Mathematical | | Instruction | Avg. |
|---|---|---|---|---|---|---|---|---|---|---|---|
| | | MME | MMB | SEED | Science QA | POPE | Hallus | Math-V | MathVista | MIA | |
| InternVL3-8B | | *Upper Bound: 100% Tokens (100%)* | | | | | | | | | |
| Vanilla | | 86.44 | 85.80 | 69.52 | 98.07 | 90.33 | 49.80 | 69.70 | 28.29 | 72.22 | 70.02 |
| | | *Retain Averaged 11.1% Tokens* (↓ 88.9%) | | | | | | | | | |
| ✗ | **Random** | 72.97 | 72.00 | 52.00 | 86.00 | 86.61 | 34.47 | 44.90 | 24.67 | 71.97 | 60.62 |
| | | ↓15.6% | ↓16.1% | ↓25.2% | ↓12.2% | ↓4.1% | ↓30.8% | ↓35.6% | ↓12.8% | ↓0.3% | |
| **Random** | ✗ | 73.11 | 72.00 | 52.00 | 85.00 | 86.32 | 34.72 | 44.00 | 21.38 | 77.93 | 60.72 |
| | | ↓15.4% | ↓16.1% | ↓25.2% | ↓13.3% | ↓4.4% | ↓30.3% | ↓36.9% | ↓24.4% | ↑7.8% | |
| ✗ | **FastV** | 76.49 | 80.00 | 54.00 | 89.00 | 88.00 | 34.88 | 47.00 | 22.04 | 68.97 | 62.26 |
| | | ↓11.5% | ↓6.8% | ↓22.3% | ↓9.2% | ↓2.6% | ↓30.0% | ↓32.6% | ↓22.1% | ↓4.5% | |
| **GPrune** | ✗ | 70.82 | 71.00 | 55.00 | 88.00 | 85.35 | 36.68 | 47.90 | **26.32** | **88.71** | 63.31 |
| | | ↓18.1% | ↓17.3% | ↓20.9% | ↓10.2% | ↓5.5% | ↓26.3% | ↓31.2% | ↓7.0% | ↑22.8% | |
| **DivPrune** | ✗ | 75.79 | **81.00** | **56.00** | **90.00** | **88.95** | **38.13** | 49.20 | **26.32** | 70.97 | **64.04** |
| | | ↓12.3% | ↓5.6% | ↓19.4% | ↓8.2% | ↓1.5% | ↓23.4% | ↓29.4% | ↓7.0% | ↓1.8% | |
| **Random** | **Random** | 73.72 | 72.0 | 51.9 | 84.6 | 86.4 | 34.5 | 45.6 | 10.29 | 69.8 | 58.76 |
| | | ↓35.4% | ↓16.1% | ↓25.4% | ↓13.7% | ↓4.3% | ↓30.7% | ↓34.6% | ↓63.6% | ↓3.4% | |
| **DivPrune** | **Random** | 74.24 | 73.5 | 53.0 | 87.3 | 87.5 | 37.1 | 47.4 | 10.68 | 71.0 | 60.19 |
| | | ↓33.0% | ↓14.3% | ↓23.7% | ↓11.0% | ↓3.1% | ↓25.5% | ↓32.0% | ↓62.2% | ↓1.7% | |
| **GPrune** | **Random** | 77.57 | 74.6 | 53.4 | 89.4 | 87.6 | 37.2 | 50.3 | 10.03 | 68.7 | 60.98 |
| | | ↓32.0% | ↓13.1% | ↓23.1% | ↓8.8% | ↓3.0% | ↓25.3% | ↓27.8% | ↓64.5% | ↓4.9% | |
| **Random** | **FastV** | 77.89 | 77.7 | 54.1 | 88.0 | 87.7 | 36.1 | 48.8 | 10.49 | 70.3 | 61.23 |
| | | ↓30.8% | ↓9.4% | ↓22.2% | ↓10.2% | ↓2.9% | ↓27.5% | ↓30.0% | ↓62.9% | ↓2.6% | |
| **GPrune** | **FastV** | 76.97 | 77.4 | 53.9 | 88.2 | 87.6 | 37.4 | 48.0 | 9.96 | 71.7 | 61.24 |
| | | ↓31.6% | ↓9.8% | ↓22.5% | ↓10.0% | ↓3.0% | ↓24.9% | ↓31.1% | ↓64.8% | ↓0.7% | |
| **DivPrune** | **FastV** | **77.97** | 80.1 | 54.2 | 89.7 | 88.5 | 37.4 | **50.3** | 10.66 | 72.6 | 62.38 |
| | | ↓31.5% | ↓6.6% | ↓22.0% | ↓8.5% | ↓2.0% | ↓28.3% | ↓30.3% | ↓62.3% | ↓1.8% | |

reveals that simply chaining two existing pruning stages underperforms the stronger single-stage baseline: the best solitary Pre-LLM method (64.04%) still surpasses every 66.7% × 66.7% hybrid method, despite the latter retaining the same number of tokens. This outcome indicates that naïvely concatenating off-the-shelf pruning criteria does not guarantee additive gains. Instead, an effective combination requires a deliberate design that respects the complementary nature of each stage as well as the downstream scenario. Without such targeted orchestration, it may be that the second stage often re-discards already informative tokens, leading to sub-optimal performance even though the aggregate sparsity is unchanged.

## 5 CONCLUSION

In this paper, we introduced UniPruneBench, a unified benchmark for evaluating visual token pruning methods in large multimodal models. By systematically covering diverse datasets, model families, pruning algorithms, and system-level efficiency metrics, UniPruneBench addresses the limitations of prior fragmented and non-standardized evaluations. Our results reveal surprising trends, including the competitiveness of random pruning, the lack of a universally superior method, and the task-specific vulnerabilities of pruning strategies. These insights highlight both the challenges and opportunities for designing more effective token compression methods. We hope UniPruneBench not only facilitates fair comparison and reproducibility but also inspires future advances in efficient multimodal learning and deployment.

## ETHICS STATEMENT

Our **UniPruneBench** benchmark is designed to provide a standardized, fair, and reproducible evaluation of visual token pruning methods in large multimodal models. The benchmark itself does not generate content or make decisions, and thus poses minimal direct ethical risk. However, potential considerations include:

1. **Data sources and bias**: UniPruneBench relies on existing public datasets, which may contain inherent biases in terms of demographics, languages, or visual concepts. Users should be aware that these biases may affect model evaluation outcomes.

2. **Misuse of results:** While the benchmark aims to improve efficiency in multimodal models, it could indirectly enable faster deployment of models in sensitive applications. We encourage responsible use and adherence to ethical AI guidelines.

3. **User queries and prompts:** Models evaluated on UniPruneBench could still produce harmful or inappropriate outputs in response to malicious or unsafe queries. The benchmark does not mitigate such risks, and appropriate safeguards should be implemented by users.

Overall, UniPruneBench aims to advance research in efficient multimodal modeling in a safe and responsible manner, providing transparency and reproducibility while minimizing ethical concerns.

## REPRODUCIBILITY STATEMENT

We provide full details to ensure that all experiments in this paper are reproducible. The code of UniPruneBench is shown in the appendix, including standardized evaluation scripts and token pruning implementations. All datasets used are publicly available, and we specify dataset preprocessing, prompt templates, token retention ratios, and system-level measurement protocols. Additionally, we include instructions for reproducing results across different model families (LLaVA, Intern-VL, and Qwen-VL) and pruning methods. By releasing the benchmark and evaluation pipeline, we aim to enable fair comparison, facilitate further research, and ensure transparency in reported findings.

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

## A    THE USE OF LARGE LANGUAGE MODELS

Large language models (LLMs) have increasingly become valuable tools for academic writing and manuscript preparation. In this work, we leverage LLMs primarily for **text refinement, language polishing, and structural editing** of our paper. This includes improving clarity, correcting grammatical errors, rephrasing sentences for conciseness, and ensuring logical flow across sections. Importantly, LLMs are used only as assistive tools. All scientific content, experiments, and analyses are independently designed, implemented, and verified by the authors. We emphasize that LLMs do not contribute to the experimental results, numerical analyses, or core intellectual content of this work. This responsible usage ensures the integrity and reproducibility of our research while benefiting from advanced language capabilities to improve presentation quality.

## B    MORE IMPLEMENTATION DETAILS

**System Configuration** Our codebase was implemented in Python 3.12 with PyTorch 2.5.1, Transformers 4.54.0 and CUDA 12.4. All experiments were conducted on NVIDIA A100-40 GB GPUs.

**Model Configuration** We evaluate three representative LMM families: LLaVA-v1.5 (7B), Qwen2.5-VL (3B & 7B), and InternVL-3 (1B & 8B), all in their official HuggingFace checkpoints without fine-tuning or weight modification. For Qwen2.5-VL, we adopt the image pre-processing setting `min_pixels=256×28×28` and `max_pixels=1280×28×28`, These settings help the model maintain image quality while controlling computational cost and resource consumption.

**Method Implementation** Pre-LLM pruning is inserted immediately after the ViT forward: visual features are collected, scored by the selected algorithm (Random-Pre, GPrune, DivPrune, etc. ), and the kept indices are used to rebuild a shorter multimodal embedding tensor before the LLM sees any tokens. Intra-LLM pruning is implemented as a per-layer hook that activates at layer $K = 2$; hidden states are split into system, visual and instruction segments, the visual subset is pruned, and position_ids, position_embeddings, and the causal mask are truncated to match the reduced token count; the subsequent layer thus computes keys/values only for the kept subset. Attention weights required by attention-based methods are obtained with a single eager-mode forward pass, saved to a temporary file, and loaded by the prune routine, keeping the modification orthogonal to Flash-Attention or SDPA code paths. All pruning decisions are executed after vision encoding and before KV-cache construction, ensuring that generation length and memory footprint shrink proportionally.

**Metric Calculation** Task scores are produced by the official VLMEvalKit evaluation scripts. For MME, we normalise the original counts to a 0–100 scale to align with other datasets; higher values indicate better performance. We further compute relative performance, allowing direct comparison of accuracy retention across tasks and pruning strengths.

**Time Measurement** Wall-clock latency is decomposed into three nested intervals: (1) *total*—end-to-end elapsed time for completing the entire benchmark; (2) *prefill*—the compute-bound encoder forward pass that processes all visual and textual tokens before the first decode step; (3) *method*—the GPU milliseconds consumed inside prefill by the pruning subroutine (token scoring, selection and tensor re-layout). Intervals are recorded with on an A100-40 GB, batch size = 1, and averaged over three runs.

## C    MORE RESULTS

We benchmark the pruning performance of existing methods on InternVL3-1B and Qwen2.5-VL-3B, with results summarized in Table 6 and Table 7, respectively. Across both model families, we observe consistent trends that align with the broader empirical patterns reported in Section Experiments. These findings reinforce the generality of our benchmark conclusions, indicating that the relative strengths and limitations of current pruning techniques are largely preserved across architectures of varying scales and designs. Such consistency underscores the reliability of our evaluation protocol and highlights the transferable insights that can be drawn from the benchmark results. Figure 4 visualizes qualitative results of token-retention mask overlays on LLaVA-v1.5-7B and LLaVA-v1.5-13B.

xcolor

Table 6: Performance comparison across different methods and benchmarks on InternVL3-1B.

| Methods | Comprehensive | | OCR | Multidisciplinary | Hallucination | | Mathematical | | Instruction | Avg. |
|---|---|---|---|---|---|---|---|---|---|---|
| | MME | MMB-en | SEED | Science QA | POPE | Hallus | Math-V | MathVista | MIA | |
| **InternVL3-1B** | *Upper Bound, 100% Tokens* **(100%)** | | | | | | | | | |
| Vanilla | 68.41 | 73.25 | 58.41 | 91.57 | 89.57 | 36.13 | 46.20 | 18.75 | 63.48 | 60.64 |
| | *Retain Averaged 33.3% Tokens* (↓ **66.7**%) | | | | | | | | | |
| **Random-Pre** | 65.06 | 65.45 | 47.30 | 83.14 | 87.39 | 32.11 | 37.40 | 14.14 | 59.85 | 54.65 |
| | ↓4.9% | ↓10.6% | ↓19.0% | ↓9.2% | ↓2.4% | ↓11.1% | ↓19.0% | ↓24.6% | ↓5.7% | |
| **Random-Intra** | 62.66 | 35.24 | 52.14 | 87.12 | 87.09 | 35.33 | 37.00 | 16.78 | 41.88 | 50.58 |
| | ↓8.4% | ↓51.9% | ↓10.7% | ↓4.9% | ↓2.8% | ↓2.2% | ↓19.9% | ↓10.5% | ↓34.0% | |
| **FastV** | 66.52 | 34.20 | 45.19 | 88.65 | 88.67 | 34.04 | 30.50 | 13.49 | 38.98 | 48.91 |
| | ↓2.8% | ↓53.3% | ↓22.6% | ↓3.2% | ↓1.0% | ↓5.8% | ↓34.0% | ↓28.1% | ↓38.6% | |
| **GPrune** | **65.78** | 71.77 | 49.56 | **89.74** | 88.27 | 33.93 | 40.80 | **18.42** | 62.71 | **57.89** |
| | ↓3.8% | ↓2.0% | ↓15.1% | ↓2.0% | ↓1.4% | ↓6.1% | ↓11.7% | ↓1.8% | ↓1.2% | |
| **DivPrune** | 66.71 | **71.26** | **52.61** | 88.00 | 88.36 | 33.13 | **42.20** | 14.14 | 61.38 | 57.53 |
| | ↓2.5% | ↓2.7% | ↓9.9% | ↓3.9% | ↓1.3% | ↓8.3% | ↓8.7% | ↓24.6% | ↓3.3% | |
| | *Retain Averaged 22.2% Tokens* (↓ **77.8**%) | | | | | | | | | |
| **Random-Pre** | 55.84 | 63.20 | 44.93 | 81.61 | 85.60 | 26.61 | 35.70 | 14.14 | 60.70 | 52.04 |
| | ↓18.4% | ↓13.7% | ↓23.1% | ↓10.9% | ↓4.4% | ↓26.4% | ↓22.7% | ↓24.6% | ↓4.4% | |
| **Random-Intra** | 59.41 | 27.45 | 43.72 | 82.42 | 86.12 | 33.55 | 32.40 | **16.78** | 46.67 | 47.62 |
| | ↓13.2% | ↓62.5% | ↓25.1% | ↓10.0% | ↓3.8% | ↓7.1% | ↓29.9% | ↓10.5% | ↓26.5% | |
| **FastV** | 64.55 | 24.24 | 42.15 | 85.73 | 87.55 | **34.40** | 31.50 | 13.49 | 41.21 | 47.20 |
| | ↓5.6% | ↓66.9% | ↓27.8% | ↓6.4% | ↓2.2% | ↓4.8% | ↓31.8% | ↓28.1% | ↓35.1% | |
| **GPrune** | 56.09 | 71.08 | 46.60 | **88.35** | 85.43 | 28.54 | **38.10** | 13.16 | 61.34 | 54.30 |
| | ↓18.0% | ↓3.0% | ↓20.2% | ↓3.5% | ↓4.6% | ↓21.0% | ↓17.5% | ↓29.8% | ↓3.4% | |
| **DivPrune** | **56.93** | 70.13 | 49.28 | 86.66 | **87.63** | 28.77 | 37.90 | 14.80 | **63.68** | **55.09** |
| | ↓16.8% | ↓4.2% | ↓15.6% | ↓5.4% | ↓2.1% | ↓20.4% | ↓17.9% | ↓21.1% | ↑0.3% | |
| | *Retain Averaged 11.1% Tokens* (↓ **88.9**%) | | | | | | | | | |
| **Random-Pre** | 52.63 | 58.70 | 41.46 | 79.52 | 82.51 | 25.79 | 33.20 | 15.46 | **61.12** | 50.04 |
| | ↓23.1% | ↓19.8% | ↓29.0% | ↓13.1% | ↓7.8% | ↓28.6% | ↓28.1% | ↓17.5% | ↓3.7% | |
| **Random-Intra** | 55.11 | 17.06 | 39.87 | 80.97 | 80.65 | 29.11 | 29.30 | 17.76 | 42.61 | 43.60 |
| | ↓19.5% | ↓76.7% | ↓31.7% | ↓11.6% | ↓9.9% | ↓19.4% | ↓36.6% | ↓5.3% | ↓32.8% | |
| **FastV** | **59.33** | 19.74 | 38.46 | **84.42** | 82.84 | **31.90** | 31.60 | 18.09 | 39.26 | 45.07 |
| | ↓13.3% | ↓73.0% | ↓34.1% | ↓7.8% | ↓7.5% | ↓11.7% | ↓31.6% | ↓3.5% | ↓38.1% | |
| **GPrune** | 55.18 | **66.32** | 42.95 | 83.74 | 75.56 | 27.09 | **33.70** | 11.51 | 61.84 | 50.88 |
| | ↓19.4% | ↓9.5% | ↓26.4% | ↓8.5% | ↓15.6% | ↓25.0% | ↓27.1% | ↓38.6% | ↓2.5% | |
| **DivPrune** | 55.41 | 65.80 | **44.53** | 83.39 | **84.58** | 26.72 | **36.30** | 11.84 | 60.68 | **52.14** |
| | ↓19.0% | ↓10.2% | ↓23.8% | ↓8.9% | ↓5.6% | ↓26.0% | ↓21.4% | ↓36.8% | ↓4.4% | |

# D DEEPER MECHANISTIC ANALYSIS OF VISUAL TOKEN PRUNING

This section provides a deeper mechanistic analysis to complement the empirical findings of UNIPRUNEBENCH, addressing the underlying reasons for the observed performance differences across various pruning strategies, task types, and model architectures.

## D.1 THE ROLE OF TOKEN REDUNDANCY AND RANDOM PRUNING EFFECTIVENESS

The empirical success of random pruning, particularly in general perception tasks, primarily stems from the **high inherent redundancy of visual tokens** generated by Vision Transformers (ViTs) and the sparse reliance of Large Multimodal Models (LMMs) on visual evidence.

1. **High Redundancy:** For many general understanding tasks, the LMM only requires a small, high-quality subset of visual features. The majority of tokens are redundant. Randomly dropping a large percentage of these tokens often preserves enough semantic structure for robust model function.

2. **Task Dependence:** This phenomenon is **strictly task-dependent**. On fine-grained tasks such as OCR or mathematical expression parsing, where **precise spatial structure and**

Table 7: Performance comparison across different methods and benchmarks on Qwen2.5-VL-3B.

| Methods | Comprehensive | | OCR | | Multidisciplinary | Hallucination | | Mathematical | | Instruction | Avg. |
|---|---|---|---|---|---|---|---|---|---|---|---|
| | MME | MMB-en | SEED | OCR-B | Science QA | POPE | Hallus | Math-V | MathVista | MIA | |
| **Qwen2.5-VL-3B** | | | | | *Upper Bound: 100% Token (100%)* | | | | | | |
| Vanilla | 77.0 | 79.2 | 68.0 | 82.3 | 81.0 | 86.7 | 45.3 | 9.9 | 38.5 | 73.5 | 64.1 |
| | | | | | *Retain Averaged 33.3% Tokens (↓66.7%)* | | | | | | |
| **GPrune** | 70.6 | 72.1 | 59.0 | 54.6 | 80.0 | 83.7 | 37.8 | 9.4 | 34.9 | 73.0 | 57.5 |
| | ↓8.3% | ↓9.0% | ↓13.2% | ↓33.7% | ↓1.2% | ↓3.4% | ↓16.6% | ↓5.1% | ↓9.4% | ↓0.7% | |
| **Random-Intra** | 70.3 | 73.5 | 56.0 | 55.5 | 80.0 | 84.3 | 36.7 | 9.0 | 37.5 | 72.9 | 57.6 |
| | ↓8.7% | ↓7.2% | ↓17.6% | ↓32.6% | ↓1.2% | ↓2.8% | ↓19.0% | ↓9.1% | ↓2.6% | ↓0.8% | |
| **Random-Pre** | 72.3 | 71.7 | 57.0 | 55.0 | 79.0 | 83.6 | 35.6 | 9.6 | 37.5 | 72.2 | 57.4 |
| | ↓6.1% | ↓9.5% | ↓16.2% | ↓33.2% | ↓2.5% | ↓3.6% | ↓21.4% | ↓3.0% | ↓2.6% | ↓1.8% | |
| **DART** | 72.0 | 75.6 | 53.0 | 56.0 | 82.0 | 80.0 | 38.1 | 9.0 | 38.5 | 73.5 | 57.8 |
| | ↓6.5% | ↓4.5% | ↓22.1% | ↓32.0% | ↑1.2% | ↓7.7% | ↓15.9% | ↓9.1% | ↑0.0% | ↑0.0% | |
| **FastV** | 70.7 | 75.3 | 57.0 | 50.7 | 80.0 | 85.4 | 40.3 | 9.7 | 38.6 | 70.4 | 57.8 |
| | ↓8.2% | ↓4.9% | ↓16.2% | ↓38.4% | ↓1.2% | ↓1.5% | ↓11.0% | ↓2.0% | ↑0.3% | ↓4.2% | |
| **DivPrune** | **72.6** | **75.4** | **60.0** | **64.9** | **80.0** | **86.4** | **39.7** | 9.1 | **39.7** | 72.5 | **60.0** |
| | ↓5.7% | ↓4.8% | ↓11.8% | ↓21.1% | ↓1.2% | ↓0.4% | ↓12.4% | ↓8.1% | ↑3.1% | ↓1.4% | |
| | | | | | *Retain Averaged 22.2% Tokens (↓77.8%)* | | | | | | |
| **GPrune** | 41.9 | 64.7 | 54.0 | 38.5 | 78.0 | 79.1 | 33.8 | **9.8** | 33.8 | 73.3 | 50.7 |
| | ↓45.6% | ↓18.3% | ↓20.6% | ↓53.2% | ↓3.7% | ↓8.7% | ↓25.4% | ↓1.0% | ↓12.2% | ↓0.3% | |
| **Random-Intra** | 65.2 | 70.3 | 53.0 | 43.7 | 80.0 | 82.3 | 36.2 | 9.0 | 36.2 | 72.9 | 54.9 |
| | ↓15.3% | ↓11.2% | ↓22.1% | ↓46.9% | ↓1.2% | ↓5.0% | ↓20.1% | ↓9.1% | ↓6.0% | ↓0.8% | |
| **Random-Pre** | 69.1 | 70.2 | 53.0 | 44.3 | 79.0 | 81.7 | 33.2 | 9.6 | 36.9 | 72.3 | 54.9 |
| | ↓10.3% | ↓11.4% | ↓22.1% | ↓46.2% | ↓2.5% | ↓5.8% | ↓26.7% | ↓3.0% | ↓4.2% | ↓1.6% | |
| **DART** | 68.9 | **74.4** | 50.0 | 48.3 | **82.0** | 76.1 | **37.2** | 9.0 | 38.0 | 72.6 | 55.7 |
| | ↓10.5% | ↓6.1% | ↓26.5% | ↓41.3% | ↑1.2% | ↓12.2% | ↓17.9% | ↓9.1% | ↓1.3% | ↓1.2% | |
| **FastV** | 68.2 | 71.5 | 53.0 | 36.3 | 80.0 | 82.6 | 37.8 | 9.4 | 37.8 | 71.7 | 54.8 |
| | ↓11.4% | ↓9.7% | ↓22.1% | ↓55.9% | ↓1.2% | ↓4.7% | ↓16.6% | ↓5.1% | ↓1.8% | ↓2.4% | |
| **DivPrune** | **69.3** | 73.9 | **57.0** | **57.5** | 80.0 | **85.3** | 36.9 | 9.2 | **39.1** | **73.9** | **58.2** |
| | ↓10.0% | ↓6.7% | ↓16.2% | ↓30.1% | ↓1.2% | ↓1.6% | ↓18.5% | ↓7.1% | ↑1.6% | ↑0.5% | |
| | | | | | *Retain Averaged 11.1% Tokens (↓88.9%)* | | | | | | |
| **GPrune** | 20.6 | 52.6 | 49.0 | 14.5 | 75.0 | 68.6 | 27.9 | 8.9 | 27.0 | 68.3 | 41.3 |
| | ↓73.2% | ↓33.6% | ↓27.9% | ↓82.4% | ↓7.4% | ↓20.8% | ↓38.4% | ↓10.1% | ↓29.9% | ↓7.1% | |
| **Random-Intra** | 60.3 | 65.5 | 48.0 | 30.8 | 77.0 | 77.2 | 31.2 | 8.8 | 34.8 | 72.0 | 50.6 |
| | ↓21.7% | ↓17.3% | ↓29.4% | ↓62.6% | ↓4.9% | ↓10.9% | ↓31.1% | ↓11.1% | ↓9.6% | ↓2.0% | |
| **Random-Pre** | 62.6 | 65.1 | 48.0 | 30.5 | 77.0 | 76.0 | 29.0 | 9.3 | 35.5 | 72.2 | 50.5 |
| | ↓18.7% | ↓17.8% | ↓29.4% | ↓62.9% | ↓4.9% | ↓12.3% | ↓36.0% | ↓6.1% | ↓7.8% | ↓1.8% | |
| **DART** | 62.0 | **69.9** | 47.0 | 29.5 | **81.0** | 68.0 | **33.1** | **9.7** | 33.1 | 73.1 | 50.6 |
| | ↓19.5% | ↓11.7% | ↓30.9% | ↓64.2% | ↑0.0% | ↓21.5% | ↓26.9% | ↓2.0% | ↓14.0% | ↓0.5% | |
| **FastV** | 20.6 | 56.6 | 49.0 | 18.6 | 79.0 | 72.1 | 33.2 | 9.0 | 31.2 | 69.0 | 43.8 |
| | ↓73.2% | ↓28.5% | ↓27.9% | ↓77.4% | ↓2.5% | ↓16.8% | ↓26.7% | ↓9.0% | ↓19.0% | ↓6.1% | |
| **DivPrune** | **63.3** | 70.2 | **52.0** | **43.1** | 78.0 | **81.7** | 34.8 | 9.0 | **36.9** | **75.0** | **54.4** |
| | ↓17.8% | ↓11.4% | ↓23.5% | ↓47.6% | ↓3.7% | ↓5.8% | ↓23.2% | ↓9.1% | ↓4.2% | ↑2.0% | |

Figure 4: Qualitative results of token-retention mask overlays on LLaVA-1.5-7B and on LLaVA-1.5-13B.

**local visual cues** are critical, random pruning exhibits sharp and consistent degradation. This confirms that random pruning, while effective for coarse redundancy removal, is fundamentally limited for detail-sensitive tasks.

## D.2   Pruning Stage and Information Flow ($ViT$-only vs. $LLM$-only)

The choice of pruning stage—before or during cross-attention—significantly impacts performance, explaining why ViT-only pruning often outperforms LLM-only pruning on certain architectures.

1. **ViT-only Pruning (Pre-modulation):** Pruning at the raw ViT feature stage removes redundancy while preserving the full **high-fidelity embedding space** before the visual tokens are modulated by cross-attention. This maintains the fidelity of critical visual information.

2. **LLM-only Pruning (Post-modulation):** Pruning after the LLM has processed the visual tokens may inadvertently remove tokens that have become **query-conditioned or attention-amplified**. If the cross-attention mechanism has not perfectly highlighted all semantically critical regions, subsequent pruning risks removing complementary tokens, potentially making the process more harmful.

## D.3   Task Sensitivity and Robustness Profiles

The observed divergent robustness profiles across capability dimensions are directly explained by the inherent visual dependence and language prior strength of the task.

1. **Vulnerability (e.g., OCR, Math Parsing):** These tasks depend on **precise local visual cues** and **fine-grained spatial structures**. Token loss in these areas directly compromises the required precision, making these tasks highly vulnerable to token pruning.

2. **Robustness (e.g., Instruction-Following):** These tasks are more robust to coarse-grained visual summaries due to stronger **language priors** and less reliance on token-level spatial patterns. The LMM can often infer the correct response based on the text prompt and a general understanding of the image.

## D.4   Deep Dive into Method-Specific Behavior

Different pruning strategies show distinct behavior depending on the pruning ratio $\rho$ and architectural constraints.

1. **Importance-Based Methods (e.g., SparseVLM) under Light Pruning** ($1 - \rho \ll 1$)**:** Methods relying on learned importance estimators (e.g., magnitude scores or attention strength) excel under light pruning. They effectively remove truly redundant tokens while retaining all semantically critical regions.

2. **Diversity-Based Methods (e.g., DivPrune) under Heavy Pruning** ($1 - \rho \approx 1$)**:** Diversity-preserving methods, which focus on maintaining **spatial coverage and feature heterogeneity**, are more stable under heavy pruning. Importance-score methods risk collapsing onto a few highly attended regions, losing complementary cues. Diversity-based methods maintain a **representative global token set**, retaining holistic scene structure and enabling superior robustness under extreme compression.

3. **Architecture–Method Interactions:** The efficacy is influenced by LMM architecture details, such as fusion depth and attention distribution sharpness. Architectures with sharper cross-attention distributions benefit more from importance-driven pruning under light compression, while those with dispersed attention may favor diversity-based methods. This necessitates careful method selection based on the target LMM family.