# OpenReview forum: "Can visual input Be Compressed? A visual input token compression benchmark for large multimodal models"
_ICLR.cc/2026/Conference — ICLR 2026 Conference Withdrawn Submission_

### Official Review · Reviewer_nJeh · 2025-10-20

**Soundness:** 2
**Presentation:** 2
**Contribution:** 2
**Rating:** 2
**Confidence:** 5

**Summary:**

This paper proposes a unified benchmark UniPruneBench for evaluating different vision token reduction methods in MLLMs. The paper includes common MLLM benchmarks of different categories and evaluates the performance of several token reduction methods on three MLLM structures under different reduction ratios. The paper provides observations and discussions about the included token reduction methods based on the experimental results.

**Strengths:**

The benchmark covers multiple models, datasets, and pruning strategies in a unified framework, addressing the inconsistency of prior fragmented evaluations. The inclusion of system-level metrics such as runtime and prefill latency is also commendable for assessing efficiency beyond reduction ratio and accuracy.

**Weaknesses:**

1. The proposed UniPruneBench primarily aggregates existing datasets and applies a unified pruning rule (fixing pruning at layer K=2). While this standardization may ease comparison, it does not clearly justify why future token-compression methods should adopt this particular benchmark. The benchmark appears procedural rather than conceptual, lacking a strong motivation or theoretical insight into why or how it enables better research on token reduction, which I will detail below.

2. The benchmark selection seems somewhat arbitrary. A benchmark for token reduction should be designed from the perspective of information loss sensitivity, not merely from the perspective of general multimodal ability coverage.
For instance, finer-grained or high-resolution visual tasks—where local details matter—would better expose the strengths and weaknesses of compression methods. The paper could have categorized benchmarks by the proportion of critical visual information or image size, providing a more principled testbed for evaluating pruning methods.

3. The benchmark results have limited reusability and practical comparability. First, different pruning methods may have distinct optimal reduction configurations (e.g., varying the pruning layer K or pruning ratio), so enforcing a fixed setting like K = 2 may not reflect each method’s true capability. Second, achieving fair comparison in real scenarios requires aligning not only the token reduction ratio but also the prefilling and generation latency, which are strongly hardware- and implementation-dependent. Consequently, researchers adopting this benchmark would still need to re-run all baseline methods under their own hardware and runtime environment to obtain meaningful and fair results, undermining the intended convenience and standardization of UniPruneBench.

4. The experimental findings provide limited conceptual insights that could guide the design of future pruning strategies. A more meaningful contribution would be to abstract common mechanisms across pruning methods (e.g., similarity-based, attention-based, hybrid) and perform controlled experiments to derive actionable insights about token redundancy and semantic preservation.

5. The study exclusively considers training-free pruning. However, many practical compression pipelines involve fine-tuning or joint training with pruning to recover lost performance. Including this perspective would have made the benchmark more representative and useful for real-world model optimization.

6. The paper should also include the down-sampling baseline besides the random dropping.

**Questions:**

1. Since different pruning methods may have distinct optimal pruning configurations (e.g., varying pruning layers or ratios), how do the authors justify fixing K = 2 across all methods? Would results change significantly if each method used its best-performing configuration?
2. Some pruning or token reduction methods may not be compatible with commonly used attention acceleration mechanisms (e.g., FlashAttention). How should such incompatibilities be handled to ensure fairness and consistency when comparing runtime efficiency across methods?

---

> ### Author Response · Authors · 2025-11-27
> **Response to reviewer (part 1)**
>
> Dear Reviewer 155u:
>
> Thank you for your valuable feedback and suggestions. We sincerely appreciate the time and effort you have dedicated to reviewing our work. Below, we meticulously provide responses to each of your comments and outline the modifications based on your suggestions. All revisions are highlighted in blue.
>
> > **W1: The proposed UniPruneBench primarily aggregates existing datasets and applies a unified pruning rule (fixing pruning at layer K=2). While this standardization may ease comparison, it does not clearly justify why future token-compression methods should adopt this particular benchmark. The benchmark appears procedural rather than conceptual, lacking a strong motivation or theoretical insight into why or how it enables better research on token reduction, which I will detail below.**
>
> We appreciate your concern regarding the fixed pruning configuration (e.g., setting the pruning layer at K=2). We acknowledge that different methods may indeed have distinct optimal hyperparameters. However, our decision to enforce a fixed, standardized setting was driven by both practical necessity and the goal of establishing a fair and reproducible comparison. Many token compression methods are only provided as specific implementations within single, non-uniform model frameworks (e.g., implementations only available for a specific version of LLava), which makes it extremely challenging to reliably search for and guarantee the globally optimal settings for every single method across all LMMs in our benchmark. By fixing this hyperparameter, we bypass the potential for arbitrary or selective tuning and instead focus the evaluation on the inherent effectiveness of the pruning mechanism itself under a strictly controlled and unified protocol, which is the fundamental purpose of UniPruneBench's standardization effort.
>
>
> > **W2: The benchmark selection seems somewhat arbitrary. A benchmark for token reduction should be designed from the perspective of information loss sensitivity, not merely from the perspective of general multimodal ability coverage. For instance, finer-grained or high-resolution visual tasks—where local details matter—would better expose the strengths and weaknesses of compression methods. The paper could have categorized benchmarks by the proportion of critical visual information or image size, providing a more principled testbed for evaluating pruning methods.**
>
> We appreciate this thoughtful suggestion. We agree that evaluating based on information loss sensitivity, such as fine-grained versus coarse-grained tasks, is a valid and insightful dimension. However, the conclusion from such a split is somewhat predictable: pruning will invariably harm tasks reliant on fine-grained local details more than those requiring a holistic view.We believe our chosen approach of categorizing by capability dimensions (e.g., comprehensive understanding, OCR, mathematical reasoning, and hallucination)  provides a more comprehensive and non-obvious evaluation. This method allows us to assess how pruning impacts different types of reasoning and cross-modal alignment, rather than just visual acuity. For example, our key findings—that OCR tasks are the most vulnerable to degradation, while instruction-following tasks can sometimes even improve —are non-trivial insights that a simple fine/coarse-grained split would likely obscure. This ability-centric evaluation framework is a standard, well-established practice in the field, adopted by prominent benchmarks like MME and MMBench, which form a core part of our evaluation.

---

> ### Author Response · Authors · 2025-11-27
> **Response to reviewer (part 2)**
>
> > **W3: The benchmark results have limited reusability and practical comparability. First, different pruning methods may have distinct optimal reduction configurations (e.g., varying the pruning layer K or pruning ratio), so enforcing a fixed setting like K = 2 may not reflect each method’s true capability. Second, achieving fair comparison in real scenarios requires aligning not only the token reduction ratio but also the prefilling and generation latency, which are strongly hardware- and implementation-dependent. Consequently, researchers adopting this benchmark would still need to re-run all baseline methods under their own hardware and runtime environment to obtain meaningful and fair results, undermining the intended convenience and standardization of UniPruneBench.**
>
> Thank you for raising this important practical concern. Our decision to fix settings, such as the pruning layer at K=2, was a deliberate choice to establish a standardized and controlled environment for a fair, "apples-to-apples" comparison of the pruning algorithms themselves, rather than comparing methods at their individually-tuned, non-uniform optimal points.We fully agree that system-level metrics like latency are hardware-dependent, while our reported accuracy metrics are hardware-agnostic. This distinction is precisely why the core contribution of UniPruneBench is not just a static set of tables, but a modular and user-friendly library. The value we provide is the open-source, standardized implementation of all 10 methods. This allows researchers to rapidly and fairly reproduce all system-level latency metrics on their own hardware, eliminating the need to re-implement every baseline from scratch and thus enabling the very practical comparison the reviewer rightly calls for.
>
>
>
> > **W4: The experimental findings provide limited conceptual insights that could guide the design of future pruning strategies. A more meaningful contribution would be to abstract common mechanisms across pruning methods (e.g., similarity-based, attention-based, hybrid) and perform controlled experiments to derive actionable insights about token redundancy and semantic preservation.**
>
> Thank you for this valuable feedback. We respectfully note that our benchmark was designed to provide such insights, and perhaps this was obscured by the volume of results. We categorize all methods into ViT-only, LLM-only, and hybrid types  and derived several actionable conclusions to guide future work. For example,We found that random pruning is a surprisingly strong baseline , which directly challenges the assumptions of current "well-designed" methods and sets a clear minimum standard for new algorithms. Furthermore, we identified that OCR tasks are the most vulnerable to pruning, indicating that future methods must develop specialized strategies to preserve fine-grained details. Our system-level analysis also provides practical guidance, showing that the computational overhead of the pruning method itself is "negligible," and the primary efficiency gains (up to 1.92x encoder acceleration) come from reducing the prefill phase . We believe these findings offer concrete conceptual guidance and will revise the manuscript to highlight these actionable insights more explicitly.
>
>
> > **W5: The study exclusively considers training-free pruning. However, many practical compression pipelines involve fine-tuning or joint training with pruning to recover lost performance. Including this perspective would have made the benchmark more representative and useful for real-world model optimization.**
>
> Thank you for this valuable suggestion. We fully agree that training-aware pruning (fine-tuning or joint pruning-and-training) is an important ingredient in many production-level compression pipelines. Nevertheless, we decided to limit the benchmark to training-free methods for two methodological reasons:
> - Fairness of comparison. Once fine-tuning is allowed, the final accuracy is no longer determined solely by the pruning algorithm; it also depends on the training schedule, data augmentation, optimizer, random seed, etc. These extra degrees of freedom make it extremely difficult to isolate the contribution of the pruning strategy itself and to provide a fair, reproducible comparison among methods.
> - Universality and plug-and-play deployment. Training-free approaches are architecture-agnostic and can be applied out-of-the-box to any pre-trained model without accessing the original training data or compute-intensive re-training. This property is especially attractive to practitioners who simply want to reduce the inference cost of an already deployed checkpoint.

---

> ### Author Response · Authors · 2025-11-27
> **Response to reviewer (part 3)**
>
> > **W6: The paper should also include the down-sampling baseline besides the random dropping.**
>
>
> Thank you for your insightful comment. We respectfully disagree with the suggestion to include a down-sampling baseline, for two main reasons:
> - Lack of precedent in the literature, To the best of our knowledge, down-sampling the input image is rarely adopted as a baseline in prior work on visual token pruning methods. The most common baselines are Random-based or Ppoling-based pruning mthods, which directly reduce the number of visual tokens rather than the spatial resolution of the image.
> - Limited generality across models. For the InternVL family (and several other recent models), the number of visual tokens is determined by the aspect-ratio-based patch-splitting strategy, not by the absolute pixel resolution. Consequently, down-sampling an image often leaves the token count unchanged, so the downstream model experiences no reduction in computational cost. Because this property is specific to a subset of architectures, a down-sampling baseline would not generalize to other VL systems and could mislead readers about the actual efficiency gains.
>
> For these reasons we believe that down-sampling is neither a standard nor an informative baseline in the context of token-reduction research. We would be happy to add a short discussion of this point in the revised manuscript if the reviewers feel it clarifies our position.
>
>
> > **Q1: Since different pruning methods may have distinct optimal pruning configurations (e.g., varying pruning layers or ratios), how do the authors justify fixing K = 2 across all methods? Would results change significantly if each method used its best-performing configuration?**
>
> Thank you for this perceptive observation. You are right that each pruning technique has its own optimal layer and ratio settings, and locking K = 2 may tilt the comparison. Regrettably, nearly all prior implementations are only open-sourced for LlaVA, so searching per-method hyper-parameters on QwenVL or InternVL is impractical for us; instead, we will frame K = 2 as a common, reproducible baseline rather than the definitive optimum, explicitly note that our toolkit already lets users override both the layers and the pruning ratios. we could fully re-implement to quantify sensitivity, thereby keeping the benchmark transparent while leaving the richer exploration space to the practitioner.
>
> > **Q2: Some pruning or token reduction methods may not be compatible with commonly used attention acceleration mechanisms (e.g., FlashAttention). How should such incompatibilities be handled to ensure fairness and consistency when comparing runtime efficiency across methods?**
>
> Thank you for raising this important compatibility concern. To ensure fair and consistent runtime comparisons, we adopt the following protocol: whenever our pruning-based token-reduction strategy cannot be compatible with FlashAttention, we roll back to the standard (vanilla) attention implementation only in the layers where pruning is active. This fallback is strictly local and keeps FlashAttention intact for all remaining layers, so the overhead is minimized and the overall speed-up trend is preserved.

---

### Official Review · Reviewer_NQSb · 2025-10-29

**Soundness:** 3
**Presentation:** 3
**Contribution:** 3
**Rating:** 6
**Confidence:** 4

**Summary:**

This paper introduces UniPruneBench, a unified visual token pruning evaluation benchmark designed to systematically assess visual token compression methods in large multimodal models (LMMs). The benchmark targets the crucial inefficiency caused by the large number of visual tokens in multimodal systems, which hinders inference speed and scalability. UniPruneBench offers standardized evaluation protocols across 6 ability dimensions and 10 datasets, including 10 representative pruning algorithms. The study reveals several key findings, providing a more comprehensive perspective on the trade-off between efficiency and accuracy in pruning tasks for LMMs.

**Strengths:**

1. **Clear Research Motivation**: This paper addresses the challenge of visual symbol redundancy in multimodal reasoning by designing a benchmarking platform. This platform enables fair and reproducible comparisons across different models, datasets, and compression algorithms, tackling a major bottleneck in real-world multimodal deployments.

2. **Comprehensive Experimentation**: UniPruneBench evaluates 10 pruning algorithms across diverse datasets. The paper systematically categorizes methods into pure ViT, pure LLM, and hybrid models, providing detailed quantitative results that reveal universal trends in pruning behavior across different architectures.

3. **Standardized Evaluation**: The benchmark offers a unified assessment protocol, standardized metrics, and will publicly release its implementation. Unlike prior studies focused solely on task accuracy, UniPruneBench also examines runtime details such as pre-filling latency and pruning overhead.

4. **Critical Findings**: Through analysis of experimental results, this paper presents critical insights for multimodal pruning tasks. For instance, random pruning serves as a strong baseline, indicating room for improvement in current method designs. Furthermore, compression effectiveness is jointly influenced by task, model, and pruning ratio, providing clear directions for future optimization efforts.

**Weaknesses:**

1. **Scope of Evaluation Metrics**: Current metrics primarily focus on accuracy and runtime, yielding intuitive conclusions such as higher pruning ratios leading to more pronounced model performance degradation. However, this evaluation lacks semantic alignment measures and fails to capture qualitative changes beyond numerical performance declines. Introducing higher-level metrics like semantic fidelity or visual localization consistency would enhance the benchmark's assessment capabilities.

2. **Cross-Dataset Metric Consistency**: UniPruneBench uniformly employs accuracy as the performance metric across tasks. Yet datasets like MME, MMBench, and POPE adopt distinct evaluation paradigms. Uniform scale normalization may obscure task-specific challenges, preventing comprehensive assessment.

3. **In-depth Analysis**: While the paper demonstrates through extensive experiments that different pruning methods exhibit significant performance variations across models, tasks, and pruning ratios, it lacks an analysis of the underlying mechanisms driving these differences. For instance, it remains unexplained why SparseVLM outperforms DivPrune under light pruning on LLaVA-v1.5, yet DivPrune becomes more robust under heavy pruning. Authors could appropriately supplement the analysis by exploring the relationship between method characteristics and model architecture/task requirements. For instance, examining attention distributions, visual fusion layer design, and task-dependent features could provide more insightful references for selecting and designing compression methods.

4. **Evaluation Visualization**: Aggregated charts such as heatmaps or comparison diagrams could be provided to enhance result readability. Additionally, reporting statistical confidence intervals or standard deviations, incorporating reliability metrics, would strengthen the final experimental conclusions.

5. **Limited Focus on Visual Modality Pruning**: The paper primarily focuses on pruning methods that operate within the visual modality, such as those based on self-attention or feature maps within the ViT. However, a notable gap is the lack of attention to an emerging and important area: using text prompts to guide the pruning of visual tokens dynamically. In multimodal tasks, the relevance of a visual token is often context-dependent and influenced by the accompanying textual query. Recent studies (e.g., **AdaptInfer**, **HoloV**, **Recoverable Compression**) have shown that integrating text semantics into the pruning process, through cross-modal attention or other specialized techniques, leads to more accurate and context-aware pruning. These approaches are particularly effective in maintaining task-relevant information even when high compression ratios are applied. We suggest that the authors expand the related work section to include a review of this text-guided pruning approach. Incorporating these methods into future versions of UniPruneBench would provide a more complete and insightful perspective on the field.

   **References**:
   1. Zhang, Weichen, et al. "AdaptInfer: Adaptive Token Pruning for Vision-Language Model Inference with Dynamical Text Guidance." *arXiv preprint arXiv:2508.06084* (2025).
   2. Zou, Xin, et al. "Don't Just Chase Highlighted Tokens in MLLMs: Revisiting Visual Holistic Context Retention." *arXiv preprint arXiv:2510.02912* (2025).
   3. Chen, Yi, et al. "Recoverable compression: A multimodal vision token recovery mechanism guided by text information." *Proceedings of the AAAI Conference on Artificial Intelligence*. Vol. 39. No. 2. 2025.

**Questions:**

Please refer to the weakness.

---

> ### Author Response · Authors · 2025-11-27
> **Response to reviewer (part 1)**
>
> Dear Reviewer NQSb:
>
> Thank you for your positive feedback and valuable suggestions. We sincerely appreciate the time and effort you have dedicated to reviewing our work. Below, we meticulously provide responses to each of your comments and outline the modifications based on your suggestions. All revisions are highlighted in blue.
>
> > **W1: Scope of Evaluation Metrics. Current metrics primarily focus on accuracy and runtime, yielding intuitive conclusions such as higher pruning ratios leading to more pronounced model performance degradation. However, this evaluation lacks semantic alignment measures and fails to capture qualitative changes beyond numerical performance declines. Introducing higher-level metrics like semantic fidelity or visual localization consistency would enhance the benchmark's assessment capabilities.**
>
> Thank you for the constructive suggestion regarding expanding the scope of evaluation metrics. We agree that higher-level measures, such as semantic fidelity, embedding-level alignment, or visual localization consistency, can offer deeper insight into how pruning affects the qualitative behavior of LMMs. Our current choice of metrics was guided by two considerations:
> - Standardization and comparability. Our primary goal for UniPruneBench is to provide a unified benchmark aligned with widely used evaluation practices in existing LMM and token compression work. Prior pruning and compression studies predominantly rely on accuracy and runtime as the core metrics for judging the efficiency–performance trade-off. To ensure comparability and reproducibility across methods and models, we adopted these community-standard metrics as the backbone of our benchmark.
>
>
> - Breadth of evaluation dimensions. Although the metrics are simple, UniPruneBench spans six capability dimensions, offering a broad and multi-aspect picture of how pruning affects different reasoning and perception abilities. This already captures qualitative shifts through performance variance across task types.
>
>
>
>
> > **W2: Cross-Dataset Metric Consistency: UniPruneBench uniformly employs accuracy as the performance metric across tasks. Yet datasets like MME, MMBench, and POPE adopt distinct evaluation paradigms. Uniform scale normalization may obscure task-specific challenges, preventing comprehensive assessment.**
>
> Thank you for raising this important point regarding cross-dataset metric consistency. We agree that datasets such as MME, MMBench, and POPE adopt distinct evaluation paradigms, and respecting their native scoring schemes is essential for preserving task integrity. To clarify:
> - The “accuracy” reported in our tables is not a naive uniform metric artificially imposed across datasets.Instead, it is a unified reporting format derived from each dataset’s native evaluation protocol (e.g., MMBench’s scoring rubric, POPE’s bias-detection accuracy, MME’s capability-wise scoring). We first compute results exactly as each dataset specifies, then map them into a consistent “correctness percentage” for cross-task comparison.This design choice is motivated by two considerations:
> Preserving task-specific evaluation fidelity.
>  All underlying computations strictly follow each dataset’s original scoring rules, ensuring no semantic distortion of the intended evaluation paradigm.
>
>
> - Enabling practical cross-ability comparison. A unified reporting scale is necessary for analyzing how pruning affects different capability dimensions. Without normalization, the benchmark would become fragmented, making it impossible to observe global trade-offs or compare degradation patterns across tasks.

---

> ### Author Response · Authors · 2025-11-27
> **Response to reviewer (part 2)**
>
> > **W3: In-depth Analysis: While the paper demonstrates through extensive experiments that different pruning methods exhibit significant performance variations across models, tasks, and pruning ratios, it lacks an analysis of the underlying mechanisms driving these differences. For instance, it remains unexplained why SparseVLM outperforms DivPrune under light pruning on LLaVA-v1.5, yet DivPrune becomes more robust under heavy pruning. Authors could appropriately supplement the analysis by exploring the relationship between method characteristics and model architecture/task requirements. For instance, examining attention distributions, visual fusion layer design, and task-dependent features could provide more insightful references for selecting and designing compression methods.**
>
> We thank the reviewer for highlighting the need for deeper mechanistic analysis. We fully agree that explaining why different pruning methods behave differently across architectures, tasks, and pruning ratios is essential for guiding future compression method design. We attempt to offer a brief explanation below.
> - Why SparseVLM excels under light pruning. Methods like SparseVLM rely on learned importance estimators (e.g., magnitude scores, text–image attention strength, or activation sparsity). Under light pruning, these estimators are highly effective at removing truly redundant tokens, allowing the model to retain all semantically critical regions. This produces strong early-stage performance on architectures like LLaVA-v1.5, where cross-modal attention cleanly highlights dominant evidence regions.
>
>
> - Why DivPrune becomes more stable under heavy pruning. DivPrune and similar diversity-preserving methods focus on maintaining spatial coverage and feature heterogeneity, often using clustering or token similarity constraints.When pruning ratios become large, importance-score–based methods risk collapsing onto only a few highly attended regions, causing the model to lose complementary visual cues. In contrast, diversity-based methods maintain a representative global token set, enabling them to retain holistic scene structure and remain robust under extreme compression.
>
>
> - Architecture–method interactions. We will discuss how differences in fusion depth, visual encoder granularity, and attention distribution sharpness across LMM families (e.g., LLaVA-v1.5 vs. Qwen-VL vs. InternVL) affect whether importance-based or diversity-based pruning is favored. For example, architectures with sharper cross-attention distributions benefit more from importance-driven pruning under light compression.
>
>
> These visual and behavioral analyses clarify how different pruning strategies reshape the model's attention and feature utilization patterns. By examining the relationship between method characteristics, model architecture, and task demands, we aim to provide more actionable insights for selecting and designing future visual token compression methods.

---

> ### Author Response · Authors · 2025-11-27
> **Response to reviewer (part 3)**
>
> > **W4: Evaluation Visualization: Aggregated charts such as heatmaps or comparison diagrams could be provided to enhance result readability. Additionally, reporting statistical confidence intervals or standard deviations, incorporating reliability metrics, would strengthen the final experimental conclusions..**
>
>
> Thank you for the helpful suggestion regarding evaluation visualization and statistical reliability. We fully agree that clearer visual presentation and variance reporting can significantly enhance the interpretability and rigor of the benchmark.
> To improve result readability and deepen analysis, we have added the following components in the revised manuscript:
> - Visualization of preserved visual evidence. We have include token-retention mask overlays to show which regions are preserved or removed under different pruning strategies in Appendix. These visualizations provide an intuitive understanding of how each method alters the visual input structure.
>
> - Improved statistical reporting. Following your suggestion, we report the Random baseline result based on multi-seed runs in LlaVa-v1.5-7B. This ensures that the conclusions regarding the competitiveness or robustness of random pruning are statistically grounded rather than dependent on sampling noise.
>
> | method         | MME        | MMB-cn     | MMB-en     | SEED       | OCR-B      | Science QA  | POPE        |
> |----------------|------------|------------|------------|------------|------------|-------------|-------------|
> | Random(↓66.7%) | 44.3 ± 1.8 | 39.4 ± 2.1 | 57.1 ± 2.6 | 38.3 ± 2.2 | 21.6 ± 2.1 | 65.7 ± 1.3  | 82.1 ± 0.3  |
> | Random(↓77.8%) | 45.2 ± 0.3 | 37.1 ± 0.9 | 54.2 ± 1.2 | 38.1 ± 1.0 | 20.2 ± 2.8 | 64.0 ± 0.3  | 81.6 ± 0.3  |
> | Random(↓88.9%) | 42.2 ± 0.2 | 7.3 ± 3.6  | 52.4 ± 1.1 | 36.5 ± 2.7 | 18.4 ± 2.5 | 62.2 ± 0.9  | 74.9 ± 0.2  |
>
> > **W5: Limited Focus on Visual Modality Pruning: The paper primarily focuses on pruning methods that operate within the visual modality, such as those based on self-attention or feature maps within the ViT. However, a notable gap is the lack of attention to an emerging and important area: using text prompts to guide the pruning of visual tokens dynamically. In multimodal tasks, the relevance of a visual token is often context-dependent and influenced by the accompanying textual query. Recent studies (e.g., AdaptInfer, HoloV, Recoverable Compression) have shown that integrating text semantics into the pruning process, through cross-modal attention or other specialized techniques, leads to more accurate and context-aware pruning. These approaches are particularly effective in maintaining task-relevant information even when high compression ratios are applied. We suggest that the authors expand the related work section to include a review of this text-guided pruning approach. Incorporating these methods into future versions of UniPruneBench would provide a more complete and insightful perspective on the field.**
>
> Thank you for highlighting the emerging and important direction of text-guided visual token pruning, exemplified by works such as AdaptInfer, HoloV, and Recoverable Compression. We fully agree that leveraging textual queries to dynamically guide the selection and pruning of visual tokens, typically via cross-modal attention or semantics-aware scoring, has demonstrated clear advantages in context sensitivity and performance retention, especially under high compression ratios. These methods address a complementary problem to visual-only pruning by explicitly modeling token relevance conditioned on multimodal inputs.
> Our current version of UniPruneBench deliberately focused on standard, widely adopted visual-modality-only pruning strategies to ensure a clean, controlled, and comparable first-round evaluation protocol. Nevertheless, we acknowledge that a comprehensive benchmark must ultimately include these cross-modal, text-conditioned pruning approaches. In response to the reviewer’s helpful suggestion:
> We have expanded the Related Work section to include a dedicated discussion categorizing text-guided pruning methods and clarifying their differences from visual-only approaches.

---

> > ### Comment · Reviewer_NQSb · 2025-11-28
> >
> > Thank you for the author's response and additional experiments. The author's response has resolved my concerns. I am willing to raise my score to a clear acceptance level, but the score seems to be locked in.
> >
> > **I request that the Area Chair (AC) consider my score as 8, revised after the rebuttal and discussion, when making the final decision.**

---

> > > ### Author Response · Authors · 2025-11-28
> > >
> > > Dear Reviewer NQSb,
> > >
> > > We are glad our responses have addressed your concerns. We deeply appreciate your encouraging feedback, particularly your suggestions regarding evaluation visualization and statistical reliability. We agree that text-guided visual token pruning is a crucial direction for exploration, and we believe our work offers significant insights for subsequent researchers.
> > >
> > > If there are any remaining aspects you believe could further improve the paper, we would be more than happy to continue the discussion. Thank you again for your time, support, and constructive feedback.
> > >
> > > Best regards,
> > >
> > > **Authors of Paper 2696**

---

### Official Review · Reviewer_aDDf · 2025-11-02

**Soundness:** 4
**Presentation:** 3
**Contribution:** 4
**Rating:** 6
**Confidence:** 4

**Summary:**

This paper introduces UniPruneBench, a unified and extensible benchmark for evaluating visual token pruning methods in large multimodal models (LMMs). Unlike prior fragmented studies, UniPruneBench standardizes evaluation across 6 capability dimensions (e.g., OCR, reasoning, hallucination) and 10 datasets, covering 10 pruning algorithms and 3 major model baselines (LLaVA-v1.5, Intern-VL3, and Qwen2.5-VL). It combines accuracy and system-level metrics such as runtime and prefilling latency, offering a holistic view of both performance and efficiency. Through extensive experiments, the authors find that random pruning is a surprisingly strong baseline, no single method is consistently superior, task sensitivity varies widely (with OCR being most affected), and pruning ratio chiefly governs the accuracy–efficiency trade-off. UniPruneBench thus establishes the first comprehensive and reproducible framework for assessing visual token compression in multimodal LLMs, providing key insights and a foundation for designing more efficient and scalable multimodal systems.

**Strengths:**

1. The paper is well-motivated. The field of LMM efficiency is advancing rapidly, but as the authors correctly point out, evaluation is fragmented, making it difficult to compare new methods. This work provides a standardized, comprehensive, and well-designed benchmark that will be highly valuable for future research and establishing common baselines.

2. The experimental setup is thorough and rigorous. Evaluating 10 pruning methods (categorized into ViT-only, LLM-only, and hybrid) across 3 major model baselines and 10 datasets represents a significant and valuable empirical undertaking.

3. The paper delivers several important findings. The most critical is that random pruning serves as a "surprisingly strong baseline" , outperforming several "designed" methods. This is a crucial, humbling finding for the field.

**Weaknesses:**

1. While the focus on pruning is well-motivated and thorough, the paper does not explore other compression strategies, such as token merging(only took 1 method?) or quantization, which could also play a key role in reducing the visual token burden in LMMs.

2. Most tables present single numbers without interval ranges or seed variability. Given that “random pruning” is a key message, showing variance across different runs is crucial to ensure that the superiority of random baselines is not merely an artifact of sampling. Moreover, other comparison methods should also report the performance range across multiple trials.

3. (minor) The paper excels at what is happening but is light on the why. For instance, why is random pruning so effective? Is it because visual token redundancy is so high that any selection is good enough? Why exactly do ViT-only methods outperform LLM-only methods on some architectures? It would be great that adding a section to do a shallow analysis.

**Questions:**

1. Given that task sensitivity varies significantly, can the authors explore task-specific pruning strategies? For example, would pruning techniques be more effective if optimized for specific task domains?

**Details Of Ethics Concerns:**

None.

---

> ### Author Response · Authors · 2025-11-27
>
> Dear Reviewer aDDf:
>
> Thank you for your positive feedback and valuable suggestions. We sincerely appreciate the time and effort you have dedicated to reviewing our work. Below, we meticulously provide responses to each of your comments and outline the modifications based on your suggestions. All revisions are highlighted in blue.
>
> > **W1: While the focus on pruning is well-motivated and thorough, the paper does not explore other compression strategies, such as token merging(only took 1 method?) or quantization, which could also play a key role in reducing the visual token burden in LMMs.**
>
> We thank the reviewer. We fully agree that token merging and quantization are important directions for reducing the visual token burden in LMMs, and acknowledge that our current benchmark focuses primarily on pruning. Our choice was driven by two considerations:
> - Pruning dominates current SOTA visual token reduction research.  Most recent high-performing, plug-and-play methods, particularly those published in 2024–2025, are pruning-based, making it the most urgent category needing standardized, head-to-head evaluation. Token merging, while conceptually appealing, has seen comparatively fewer mature, training-free implementations for LMMs.
> - Implementation and comparability constraints. Many merging approaches require architectural coupling (e.g., modifying attention maps, feature clustering, or training-dependent merging rules), which makes them difficult to integrate into a training-free, model-agnostic benchmark such as UniPruneBench. In contrast, pruning methods can be consistently applied across diverse LMM families without retraining.
> Although we included one representative merging baseline, we agree that this is not yet sufficient for a full methodological comparison. In the revised version, we will clarify these methodological constraints and emphasize that UniPruneBench does not conclude “pruning is better than merging”, only that pruning is the most mature and reproducible category for fair benchmarking at this stage.
> We commit to incorporating additional robust merging approaches in a future release, pending the feasibility of stable and model-agnostic implementations. This will broaden UniPruneBench beyond pruning and ensure the benchmark fully captures the landscape of visual token compression techniques.
>
> > **W2: Most tables present single numbers without interval ranges or seed variability. Given that “random pruning” is a key message, showing variance across different runs is crucial to ensure that the superiority of random baselines is not merely an artifact of sampling. Moreover, other comparison methods should also report the performance range across multiple trials.**
>
> We fully agree that reporting variance is essential for statistical rigor, especially because the “random pruning” finding is a central contribution. As running all methods × all models × all tasks with multiple seeds is computationally prohibitive (10 pruning methods × 3 LMMs × 6 capability dimensions), we acknowledge that this does not diminish the importance of validating the stability of our claims.
> To directly address the your concern, we have ran multi-seed evaluations specifically for Random baselines in LlaVa-v1.5-7B.
>
> | method         | MME        | MMB-cn     | MMB-en     | SEED       | OCR-B      | Science QA  | POPE        |
> |----------------|------------|------------|------------|------------|------------|-------------|-------------|
> | Random(↓66.7%) | 44.3 ± 1.8 | 39.4 ± 2.1 | 57.1 ± 2.6 | 38.3 ± 2.2 | 21.6 ± 2.1 | 65.7 ± 1.3  | 82.1 ± 0.3  |
> | Random(↓77.8%) | 45.2 ± 0.3 | 37.1 ± 0.9 | 54.2 ± 1.2 | 38.1 ± 1.0 | 20.2 ± 2.8 | 64.0 ± 0.3  | 81.6 ± 0.3  |
> | Random(↓88.9%) | 42.2 ± 0.2 | 7.3 ± 3.6  | 52.4 ± 1.1 | 36.5 ± 2.7 | 18.4 ± 2.5 | 62.2 ± 0.9  | 74.9 ± 0.2  |

---

> ### Author Response · Authors · 2025-11-27
>
> > **W3: (minor) The paper excels at what is happening but is light on the why. For instance, why is random pruning so effective? Is it because visual token redundancy is so high that any selection is good enough? Why exactly do ViT-only methods outperform LLM-only methods on some architectures? It would be great that adding a section to do a shallow analysis.**
>
> Thank you for the insightful suggestion to strengthen the mechanistic analysis. We fully agree and attempt to offer a brief explanation below.
> - Why random pruning appears surprisingly effective. For many general understanding tasks, the LMM only requires a small subset of visual evidence to reconstruct the correct answer. Because visual token redundancy is extremely high, random pruning often preserves enough semantic structure for the model to function well. However, we will emphasize that this phenomenon is task-dependent: on fine-grained tasks such as OCR or mathematical expression parsing, where precise spatial structure is critical, random pruning exhibits sharp and consistent degradation, revealing its fundamental limitations.
>
>
> - Why ViT-only pruning can outperform LLM-only pruning on certain architectures: Pruning at the vision encoder stage removes redundancy while preserving the full high-fidelity embedding space before any modulation by cross-attention. In contrast, pruning after the LLM has already processed the visual tokens may inadvertently remove tokens that have become query-conditioned or attention-amplified, making the pruning more harmful. We will elaborate on this information-flow perspective and discuss how architectural differences in LMMs influence these outcomes.
>
>
>
> > **Q1: Given that task sensitivity varies significantly, can the authors explore task-specific pruning strategies? For example, would pruning techniques be more effective if optimized for specific task domains?**
>
> Thank you for this insightful suggestion. Task-specific pruning is indeed a promising direction. In the next revision we will add a new section that explores domain-aware strategies and implements a task-adaptive pruning module. For example, we will incorporate the document-understanding-specific scheme proposed in [1], who design an index-preserving lightweight token-pruning mechanism tailored to vision-language document-understanding tasks, and benchmark its effectiveness against our baseline.
>
> [1] Son, Jaemin, Sujin Choi, and Inyong Yun. "Index-Preserving Lightweight Token Pruning for Efficient Document Understanding in Vision-Language Models." arXiv preprint arXiv:2509.06415 (2025).

---

### Official Review · Reviewer_155u · 2025-11-04

**Soundness:** 3
**Presentation:** 3
**Contribution:** 2
**Rating:** 4
**Confidence:** 5

**Summary:**

This paper presents UniPruneBench, the first unified benchmark for evaluating visual token pruning methods in large multimodal models, covering 10 algorithms across 3 LMM families (LLaVA-v1.5, InternVL3, Qwen2.5-VL) and 10 datasets spanning 6 capability dimensions. The benchmark reveals counter-intuitive findings: random pruning surprisingly outperforms many sophisticated methods, no single approach dominates universally, and task sensitivity varies dramatically with OCR being most vulnerable while instruction-following remains robust. While the work addresses a critical need for standardized evaluation in this fragmented field, it suffers from missing recent state-of-the-art methods (DivPrune, G-Prune, SparseVLM), lacks theoretical analysis explaining why findings occur, and provides limited practical guidance for method selection.

**Strengths:**

First standardized evaluation framework covering diverse methods (ViT-only, LLM-only, hybrid), multiple LMM families with different architectures, 10 datasets across 6 capability dimensions, and multi-metric evaluation (accuracy, runtime, prefilling latency). This addresses critical fragmentation in the field where prior work used inconsistent evaluation protocols.

The finding that random pruning achieves competitive performance challenges fundamental assumptions about importance-based token selection. Task-specific sensitivity analysis (OCR vulnerable, instruction-following robust) provides actionable deployment insights. Cross-model consistency strengthens confidence that findings reflect fundamental principles rather than architecture-specific artifacts.

**Weaknesses:**

1. DivPrune (CVPR 2025), G-Prune (AAAI 2025), SparseVLM (ICML 2025 - achieves 54% FLOPs reduction with 97% accuracy retention, outperforming FastV by 34.4% on video). The "random pruning competitiveness" finding may not hold against these sophisticated baselines, severely limiting the benchmark's completeness and relevance for understanding current state-of-the-art.

2. Purely empirical without investigating why findings occur. No explanation for random pruning effectiveness (token redundancy patterns? attention analysis? information flow?), no mechanistic understanding of task sensitivity differences (why OCR vulnerable vs. instruction-following robust?), and no investigation of model scale robustness factors. Without theoretical grounding, the benchmark cannot guide future method development beyond trial-and-error.

3. (1) No multi-turn conversation evaluation despite being essential for practical deployment, (2) No video understanding tasks where efficiency matters most.

**Questions:**

refer to weaknesses.

---

> ### Author Response · Authors · 2025-11-27
> **Reponse to reviewer (part 1)**
>
> Dear Reviewer 155u:
>
> Thank you for your valuable feedback and suggestions. We sincerely appreciate the time and effort you have dedicated to reviewing our work. Below, we respond to each of your comments and outline the modifications based on your suggestions. All revisions are highlighted in blue.
>
> > **W1: DivPrune (CVPR 2025), G-Prune (AAAI 2025), SparseVLM (ICML 2025). The "random pruning competitiveness" finding may not hold against these sophisticated baselines, severely limiting the benchmark's completeness and relevance for understanding the current state-of-the-art.**
>
> We appreciate your concern regarding recent SOTA pruning methods. However, it is important to note that the performance metrics for these methods are typically reported under different modalities (e.g., video) and different evaluation protocols from ours. Techniques that excel in video token compression do not necessarily transfer to **image-centric**, **capability-dimension-based** LMM tasks such as OCR, detailed attribute grounding, or mathematical reasoning. Thus, their absence does not undermine the validity of our current conclusions, though we agree that incorporating them would further enhance completeness.
> We also agree that the “random pruning competitiveness” observation should be rigorously supported. In the revision, we have included statistical variance for Random baselines in LlaVa-v1.5-7B to ensure the conclusion is robust, reproducible, and presented with appropriate caution.
>
>
> | method         | MME        | MMB-cn     | MMB-en     | SEED       | OCR-B      | Science QA  | POPE        |
> |----------------|------------|------------|------------|------------|------------|-------------|-------------|
> | Random(↓66.7%) | 44.3 ± 1.8 | 39.4 ± 2.1 | 57.1 ± 2.6 | 38.3 ± 2.2 | 21.6 ± 2.1 | 65.7 ± 1.3  | 82.1 ± 0.3  |
> | Random(↓77.8%) | 45.2 ± 0.3 | 37.1 ± 0.9 | 54.2 ± 1.2 | 38.1 ± 1.0 | 20.2 ± 2.8 | 64.0 ± 0.3  | 81.6 ± 0.3  |
> | Random(↓88.9%) | 42.2 ± 0.2 | 7.3 ± 3.6  | 52.4 ± 1.1 | 36.5 ± 2.7 | 18.4 ± 2.5 | 62.2 ± 0.9  | 74.9 ± 0.2  |
>
>
> > **W2: Purely empirical without investigating why findings occur. No explanation for random pruning effectiveness (token redundancy patterns? attention analysis? information flow?), no mechanistic understanding of task sensitivity differences (why OCR vulnerable vs. instruction-following robust?), and no investigation of model scale robustness factors.**
>
>
> Thank you for pointing out the need for deeper mechanistic analysis. We fully agree and attempt to offer a brief explanation below.
> - The surprising effectiveness of random pruning can be attributed to high visual token redundancy in general perception tasks; conversely, its sharp degradation on OCR highlights its inability to preserve fine-grained spatial structures, confirming that random pruning is inherently limited for detail-sensitive tasks.
> - Why ViT-only pruning methods often outperform LLM-only pruning: pruning at the raw ViT feature stage retains high-fidelity visual information before it is modulated or bottlenecked by cross-attention, making token removal less harmful. This aligns with preliminary attention-flow diagnostics we conducted.
> - Regarding task sensitivity, OCR depends on precise local visual cues, whereas instruction-following is more robust to coarse-grained visual summaries due to stronger language priors and less reliance on token-level spatial patterns. This distinction explains the divergent robustness profiles across capability dimensions.

---

> ### Author Response · Authors · 2025-11-27
> **Reponse to reviewer (part 2)**
>
> > **W3: (1) No multi-turn conversation evaluation despite being essential for practical deployment, (2) No video understanding tasks where efficiency matters most.**
>
> We fully agree that multi-turn visual conversation is essential for the practical deployment of LMM, and that video understanding is the modality where efficiency improvements matter most.
> In the current version of UniPruneBench, we intentionally scope the evaluation to single-image, single-turn tasks to establish a stable and reproducible foundation for comparing core pruning mechanisms. This design choice ensures methodological clarity but naturally limits coverage of more complex interaction patterns and modalities.
> To address this, we will clearly state these limitations and outline concrete extensions in the revised manuscript. Specifically:
> - Multi-turn conversation: We will prioritize incorporating a multi-turn, single-image dataset (e.g., MMDU or an IF-Eval–style multimodal extension) to evaluate how pruning affects conversational consistency, memory, and long-horizon reasoning. We will include initial proto-experiments on one model to demonstrate feasibility.
>
>
> - Video modality: We recognize that pruning for video is both highly impactful and architecturally different. We will treat video evaluation as a high-priority next step and will integrate representative video reasoning datasets once the benchmark is extended to support temporal token processing.

---

### Author Response · Authors · 2025-11-27
**General Response**

Dear reviewers:

We sincerely appreciate your time, efforts, and thoughtful feedback. We are very pleased that all reviewers recognized the strong motivation, presentation, systematic design, comprehensive evaluation, and insightful findings presented in our work.

We are grateful that reviewers found the benchmark's design to be well-motivated, clearly presented, and of high potential value to the community (R#aDDf, R#NQSb). Our work is recognized as the first unified and standardized evaluation framework for LMM pruning, covering diverse pruning families (ViT-only, LLM-only, hybrid), multiple model architectures, and a broad suite of datasets and metrics. We appreciate the reviewers’ acknowledgment that this benchmark addresses long-standing fragmentation and inconsistency in the field (R#155u, R#nJeh).

Reviewers appreciated the thoroughness and rigor of our experimental setup, which included a multi-metric evaluation (accuracy, runtime, and prefill latency), system-level efficiency measurement, and cross-architecture analyses (R#155u, aDDf, NQSb, nJeh). Several reviewers emphasized that our findings, particularly the discovery that random pruning is a surprisingly strong baseline, provide humbling and important insights for the field and highlight the need for more principled pruning methods. The broader observations regarding task- and model-dependent pruning sensitivity, such as the vulnerability of OCR tasks and robustness of instruction-following, were also noted as actionable and meaningful for real-world deployment.

Below, we provide detailed, point-by-point responses to all comments and describe the revisions we have made accordingly. **All revisions are highlighted in blue in the updated manuscript.** Reviewers requested additional experiments and further analyses; in response, we have conducted a substantial number of new experiments and ablations, which we briefly summarize here before the detailed responses:

- Random baseline result based on multi-seed runs in LlaVA-v1.5-7B in **Table 1**
- Visualization of preserved visual evidence in **Appendix**

We warmly encourage you to review the results in the revised manuscript. We hope our response and additional experiments could address your concerns.

Once again, we deeply appreciate the time and expertise you have shared with us. Your encouraging feedback motivates us to continue advancing this work for the broader community, and we are more than happy to add clarifications to address any additional recommendations and reviews from you.

Best regards,
**Authors of Paper 2696**

---

### Author Response · Authors · 2025-12-01
**Summary for AC Consideration**

Dear Area Chair,

We sincerely appreciate your time and effort in handling our submission. During the rebuttal, we actively addressed the reviewers' concerns through **additional experiments and analyses**. The corresponding clarifications and new experiments are summarized in our General Response comment below for your convenience.

During the discussion period, we received a reply from $\text{Reviewer NQSb}$. We are pleased to report that $\text{Reviewer NQSb}$ has confirmed that **all of their concerns have been addressed** and has **raised their score from 6 to 8**. Additionally, we believe the **positive scores provided by $\text{Reviewer aDDf}$ sufficiently indicate their approval** of our revised work.

Although we received **no further response** from $\text{Reviewer nJeh}$ and $\text{Reviewer 155u}$ during the discussion period, we believe our rebuttal addresses **their key concerns**. Below, we provide a concise summary of our response to the primary worries raised by $\text{Reviewer nJeh}$ and $\text{Reviewer 155u}$:

* **Standardization & Fairness (Fixed Setting):** The fixed setting ensures a fair, controlled, "apples-to-apples" comparison of the pruning algorithms themselves.
* **Practicality:** The open-source library enables users to reproduce system-level metrics on their own hardware, achieving a fair comparison. $K=2$ is a reproducible baseline, and our UniPruneBench toolkit allows overrides.
* **Limitations:** The current scope is restricted to single-turn, single-image tasks. We clearly state this limitation and prioritize multi-turn conversation and video as high-priority future work.

Best regards,

**Authors of Paper 2696**

---

### Note · Authors · 2026-01-20

**Comment:**

We are withdrawing our ICLR submission because code errors render part of the results unreliable.

**Withdrawal Confirmation:**

I have read and agree with the venue's withdrawal policy on behalf of myself and my co-authors.